# Modeling urban pollutant transport at multi-resolutions: Impacts of turbulent mixing

Zining Yang[1], Qiuyan Du[1], Qike Yang[1], Chun Zhao[1,2,3*], Gudongze Li[1], Zihan Xia[1], Mingyue Xu[1], Renmin Yuan[1], Yubin Li[4], Kaihui Xia[1], Jun Gu[1], and Jiawang Feng[1]

[1]Deep Space Exploration Laboratory/School of Earth and Space Sciences/CMA-USTC Laboratory of Fengyun Remote Sensing/State Key Laboratory of Fire Science/Institute of Advanced Interdisciplinary Research on High-Performance Computing Systems and Software, University of Science and Technology of China, Hefei, China

[2]Laoshan Laboratory, Qingdao, China

[3]CAS Center for Excellence in Comparative Planetology, University of Science and Technology of China, Hefei, China.

[4]School of Atmospheric Physics, Nanjing University of Information Science and Technology, Nanjing, China

*Corresponding author: Chun Zhao (chunzhao@ustc.edu.cn)

**Key Points:**

1. Higher horizontal resolutions improve BC surface concentration predictions by enhancing PBL mixing and vertical wind flux, especially at night.

2. Small-scale eddies resolved at higher horizontal resolutions strengthen vertical fluxes, increasing BC atmospheric lifetime and column concentrations.

3. **Detailed land use and terrain in high horizontal resolution models enhance PBL mixing, refining pollutant transport and urban air quality simulations.**

## Abstract

Air pollution in cities impacts public health and climate. Turbulent mixing is crucial in pollutant formation and dissipation, yet current atmospheric models struggle to accurately represent it. Turbulent mixing intensity varies with model resolution, which has rarely been analyzed. To investigate turbulent mixing variations at multi-resolutions and their implications for urban pollutant transport, we conducted experiments using WRF-Chem at 25, 5, and 1 km resolutions. The simulated meteorological fields and black carbon (BC) concentrations are compared with observations. Differences in turbulent mixing across multi-resolutions are more pronounced at night, resulting in noticeable variations in BC concentrations. BC surface concentrations decrease as resolution increases from 25 km to 5 km and further to 1 km, but are similar at 5 km and 1 km resolutions. Enhanced planetary boundary layer (PBL) mixing coefficients and vertical wind flux at higher resolutions reduce BC surface concentration overestimations. The 1 km resolution parameterized lower mixing coefficients than 5 km but resolved more small-scale eddies, leading to similar near-surface turbulent mixing at both resolutions, while the intensity at higher altitudes is greater at 1 km. This caused BC to be transported higher and farther, increasing its atmospheric lifetime and column concentrations. Variations in mixing coefficients are partly attributed to differences in land use and terrain, with higher resolutions providing more detailed information that enhances PBL mixing coefficients, while grid size remains crucial in regions with more gradual terrain and land use changes. This study interprets how turbulent mixing affects simulated urban pollutant diffusion at multi-resolutions.

## 1. Introduction

Since the middle of the 19th century, rapid economic growth and urbanization have caused severe regional haze and photochemical smog pollution (Li et al., 2015; Li et al., 2019; Ma et al., 2019). A variety of air pollution episodes mainly occur in cities (Chan and Yao, 2008). Exposure to atmospheric particulate matter is one of the major threats to public health (Yin et al., 2017; Liu et al., 2019). Accurate pollutant estimation is crucial for the realization of pollution prevention goals. Pollution processes are affected by many different factors, such as pollution source emissions (Li et al., 2017a), physical and chemical characteristics of aerosols (Riccobono et al., 2014; Zhao et al., 2018), topographic effects (Zhang et al., 2018), and meteorological conditions (Ye et al., 2016). Significantly, pollutant concentrations are mainly gathered within the planetary boundary layer (PBL), and PBL mixing processes are associated with intricate turbulent eddies (Stull, 1988), which significantly affect the horizontal transport and vertical diffusion of pollutants (Wang et al., 2018; Du et al., 2020; Ren et al., 2020; Ren et al., 2021), as well as the formation of new aerosol particles (Wu et al., 2021).

The mechanism of turbulent transport has been widely investigated. The vertical diffusion of pollutants in urban areas is affected by the structure of the urban boundary layer (UBL), and different structures may lead to uneven spatial distribution of pollutants (Han et al., 2009; Zhao et al., 2013c). First, meteorological conditions play dominant roles in turbulent mixing of air pollutants within the atmospheric boundary layer (ABL) (Xu et al., 2015; Miao et al., 2019). Unstable meteorological conditions enhance turbulence, promoting pollutant dispersion, while stable conditions suppress it, leading to pollutants accumulation. Previous studies have indicated that constant stagnant winds and increased water vapor density inhibit the vertical diffusion of pollutants, resulting in explosive growth of pollutants (Zhang et al., 2015a; Zhang et al., 2015b; Wei et al., 2018; Zhong et al., 2018). Under these stable conditions, the inherent characteristics of the stable boundary layer (SBL), particularly turbulence intermittency (Costa et al., 2011), affect the heavy urban haze events by altering surface-atmosphere exchanges (Wei et al., 2018; Ren et al., 2019a; Ren et al., 2019b; Wei et al., 2020; Ren

et al., 2021; Zhang et al., 2022). Second, diurnal variations in turbulent mixing between
day and night significantly influence changes in pollutant concentrations (Li et al., 2018;
Liu et al., 2020). In the daytime convective boundary layer (CBL), pollutants can be
mixed uniformly in a thick layer due to the intense turbulent mixing (Sun et al., 2018).
While in the nighttime SBL, reduced mixing and dispersion result in the accumulation
of pollutants near the surface (Holmes et al., 2015). Severe urban haze pollution
formation is typically accompanied with the development of nocturnal SBL (Pierce et
al., 2019; Li et al., 2020; Zhang et al., 2020; Li et al., 2022). Moreover, pollutants in
the residual layer can be mixed downward to the surface with the development of the
ABL the next morning (Chen et al., 2009; Sun et al., 2013; Quan et al., 2020). Overall,
the impact of turbulent mixing on urban pollution is important and complex.
Numerical simulation is an important method for studying turbulent mixing.
However, there are still challenges associated with accurately representing turbulent
mixing in numerical models. Previous research has indicated that turbulent mixing in
current atmospheric chemical models is insufficient to capture stable atmospheric
conditions, potentially leading to rapid increases in severe haze in urban areas (Wang
et al., 2018; Peng et al., 2018; Ren et al., 2019b; Du et al., 2020). Some studies revealed
that WRF-Chem simulations underestimate turbulent exchange within stable nocturnal
boundary layers, allowing unrealistic accumulation of pollutants near the surface
(Tuccella et al., 2012; Berger et al., 2016). Additionally, PBL parameterization schemes
in current models may not accurately represent intricate turbulent mixing, particularly
in complex terrains, urban areas, or extreme weather conditions. Research has revealed
that different PBL parameterization schemes employed in WRF-Chem tend to
underestimate turbulent mixing when compared to observations (Hong et al., 2006;
Banks and Baldasano, 2016; Kim, 2006). Turbulent mixing coefficients diagnosed in
atmospheric models characterize the intensity of turbulent mixing (Cuchiara et al.,
2014). However, these models frequently underestimate mixing coefficients during the
nighttime. Previous research has employed various approaches to address this
limitation. Du et al. (2020) demonstrated that increasing the lower limit of PBL mixing
coefficients during nighttime significantly reduced the modeling biases in simulated
pollutant concentrations. Jia and Zhang (2021) utilized the new modified turbulent
diffusion coefficient to represent the mixing process of pollutants separately and
improved the simulation results of pollutant concentrations. Jia et al. (2021) employed
the revised turbulent mixing coefficient of particles using high-resolution vertical flux
data of particles according to the mixing length theory, and improved the overestimation
of pollutant concentrations. In conclusion, current atmospheric models commonly face
several challenges in accurately simulating turbulent mixing.
The representation of turbulent mixing in models is influenced by various factors,
including grid resolution, topography, boundary layer parameterization, atmospheric
dynamics, and land-surface processes. Among these factors, model resolution can
significantly affect turbulent mixing processes in atmospheric simulations, with
simulated turbulent mixing varying substantially across different resolutions. Qian et al.
(2010) evaluated model performance at 3 km, 15 km, and 75 km resolutions, finding
that only simulations at 3 km resolution accurately captured multiple concentration
peaks in observational data, indicating that turbulent mixing may play a critical role in
simulating pollutant concentrations. Fountoukis et al. (2013) conducted model
simulations at three resolutions and demonstrated that higher resolution reduced the
bias for BC concentration by more than 30% in the Northeastern United States during
winter, attributing this improvement to better resolved pollutant dispersion. Tao et al.
(2020) found that changes in model resolution led to increased pollutant concentrations
in urban areas but decreased concentrations in west mountain regions, likely due to
differences in vertical and horizontal dispersion. In conclusion, previous research has
primarily focused on comparing pollutant concentrations across different model
resolutions, demonstrating that resolution significantly affects pollutant distribution
and dispersion. These studies suggest that turbulent mixing may play a crucial role.
However, few have systematically explored the specific mechanisms by which
turbulent mixing influences pollutant concentrations simulated at multi-resolutions,
despite their importance in determining urban atmospheric pollutions.
Motivated by aforementioned problems, this study aims to investigate differences
in pollutant concentrations across multi-resolutions and explore how the turbulent

mixing plays as a crucial role affecting pollutant concentrations at various resolutions. Furthermore, we seek to determine whether higher-resolution simulations can address the issue of inaccurate turbulent mixing in current models. The Weather Research and Forecasting model coupled with Chemistry (WRF-Chem) is applied to simulate pollutant and meteorological fields during the spring of 2019 in Hefei, a typical mega-city and sub-center of the Yangtze River Delta (YRD) urban agglomeration in China, with a population of nearly 10 million and an area of 11,445 km$^2$. Our study interprets the various characteristics of black carbon (BC) distributions simulated at multi-resolutions and focuses on the mechanisms involved. BC is selected as the primary pollutant for this study due to its near-inert nature in the atmosphere and can be treated as a representative tracer for turbulent mixing. The paper is organized as follows: Section 2 introduces the WRF-Chem model configuration, the design of multi-resolutions experiments, emissions from different sources, and observational data. Section 3 evaluates model simulations across multi-resolutions against observations, presents the spatial distributions of surface and column concentrations simulated at three resolutions, and investigates the important turbulent mixing processes that generate spatial variability in pollutant concentrations. Section 4 present the conclusion and discussion of the analysis.

## 2. Methodology

### 2.1 Models and Experiments

2.1.1 WRF-Chem

The non-hydrostatic Weather Research and Forecasting (WRF) model includes various options for dynamic cores and physical parameterizations that can be used to simulate atmospheric processes over a wide range of spatial and temporal scales (Skamarock et al., 2008). WRF-Chem, the chemistry version of the WRF model (Grell et al., 2005), simulates trace gases and particulates interactively with the meteorological fields. WRF-Chem treats photochemistry of trace gases and aerosol-related processes with various different schemes (e.g. the Statewide Air Pollution Research Center

(SAPRC99) photochemical mechanism and the Model for Simulating Aerosol Interactions and Chemistry (MOSAIC)). In this study, the version of WRF-Chem updated by the University of Science and Technology of China (USTC version of WRF-Chem) is used. Compared with the publicly released version, this USTC version of WRF-Chem includes some additional functions such as the diagnosis of radiative forcing of aerosol species, land surface coupled biogenic VOC (volatile organic compound) emission, aerosol-snow interaction, improved PBL mixing of aerosols and a detailed diagnosis of the contributions of each crucial process to pollutant concentrations (Zhao et al., 2013a; Zhao et al., 2013b; Zhao et al., 2014; Zhao et al., 2016; Hu et al., 2019; Du et al., 2020; Zhang et al., 2021).

The configuration of WRF-Chem in this study is given in Table 1. The SAPRC99 photochemical mechanism (Carter, 2000) is chosen to simulate the gas-phase chemistry, and the MOSAIC is selected for aerosol processes (Zaveri and Peters, 1999; Zaveri et al., 2008). The MOSAIC aerosol scheme includes important physical and chemical processes such as nucleation, condensation, coagulation, aqueous-phase chemistry, and water uptake by aerosols. Sulfate, nitrate, ammonium, sea salt, mineral dust, organic matter (OM), BC, and other (unspecified) inorganics (OIN) constitute the prognostic species in MOSAIC. The aerosol direct effect is coupled to the Rapid Radiative Transfer Model (RRTMG) (Mlawer et al., 1997; Iacono et al., 2000) for both SW (shortwave) and LW (longwave) radiation as implemented by Zhao et al. (2011). We also turned on the aerosol indirect effect, which represents the interactions between aerosols and clouds, including the first and second indirect effects, activation/resuspension, wet scavenging, and aqueous chemistry (Gustafson et al., 2007; Chapman et al., 2009). The photolysis rate is computed by the Fast-J radiation parameterization (Wild et al., 2000). Our simulation includes the secondary organic aerosol (SOA) mechanism, a crucial aerosol process that can substantially reduce discrepancies between simulated results and observations.

Another type of option is meteorological physics, including the Yonsei University (YSU) nonlocal PBL parameterization scheme (Hong et al., 2006), the Noah land-surface model (Chen and Dudhia, 2001) for the surface layer process, the Morrison

two-moment scheme (Morrison et al., 2009) for cloud microphysics, and the Rapid
Radiative Transfer Model (RRTMG) for longwave and shortwave radiation. The 25 km
resolution simulation turns on the option of cumulus parameterization, which uses the
Kain-Fritsch cumulus and shallow convection scheme (Kain, 2004) to simulate sub-
grid scale clouds and precipitation. However, this option is turned off in the other two
higher resolution simulations because the fine-resolution is sufficient to resolve the
cloud forming processes.

2.1.2 Numerical experiments
The study period spans from March 5th to March 20th, 2019. Following previous
research (Gustafson et al., 2011), the first five days are considered as the model spin-
up time, while the remaining integration period is used for analysis. Consequently, only
the results from March 10th to March 20th, 2019, are used in the analysis of this study.
Three different resolutions and computational domains are employed in our study. The
outer domain, covering East, North, and South China, has 140 x 105 grid cells (107.1°-
127.9°E, 17.1°-44.9°N) with a horizontal resolution of 25 km. The middle domain,
encompassing the entire YRD region in East China, has 250 x 250 grid cells (111.82°-
121.78°E, 27.02°-36.98°N) with a resolution of 5 km. The inner domain, covering most
of the Hefei region, consists of 150 x 150 grid cells (116.604°-117.796°E, 31.204°-
32.396°N) at a horizontal resolution of 1 km. The center of inner domain is the city of
Hefei, a typical mega-city of East China. Hefei, the capital city of Anhui province, is
located in the mid-latitude zone with a humid subtropical monsoon climate and serves
as a representative case for this study. The regions are shown in Figure S1. To facilitate
the comparison of discrepancies among the three simulations at different resolutions,
we have selected the innermost region as the main scope of study for this research, as
shown in Figure 1a.
In this study, we derive terrain information from a high-resolution (~ 1 km) US
Geological Survey (USGS) topographic data and interpolate it onto the WRF grid.
Therefore, the three domains with different resolutions exhibit varying degrees of
terrain detail. The 1 km grid resolves the most intricate topographic features, followed
by the 5 km grid, while the 25 km grid captures the least spatial detail. These multi-
resolutions topographic representations potentially influence pollutant turbulent mixing
processes, which will be analyzed in this study. The land cover dataset is derived from
a 1 km horizontal resolution dataset for China (Zhang et al., 2021). The land use
categories follow the United States Geological Survey's (USGS) 24-category
classification, and the dataset is based on China's land cover conditions as of 2015. This
provides a more accurate representation of current land cover, particularly for eastern
China, which has experienced intensive urban expansion since the 2000s. Figure 1b
shows the land cover data at different resolutions, with detailed descriptions of the
legend and land cover classes provided in Table S1. This set of simulations is referred
to as the baseline experiment. With the exception of part of Section 3.2.3, all other
analyses in this study are based on the results of these baseline experiments. Moreover,
to explore the differences in turbulent mixing simulated at multi-resolutions under
consistent land use conditions, we conducted an additional set of sensitivity
experiments referred to as the sensitivity experiment. The sensitivity experiment was
identical to the baseline experiment, except it used the default USGS land use category
data in WRF. Notably, this default USGS data in WRF's geographical static database
represents Chinese land use patterns before the 2000s, as shown in Figure S2. This
default dataset reflects land use distribution prior to China's significant urbanization.
Consequently, the land use data types have minor variations and remained generally
consistent across all three resolutions in the sensitivity experiment.
On the other hand, the vertical configuration within the PBL is also crucial for
accurately modeling pollutant dispersion. To better resolve the PBL structure and
mixing processes, we implemented a finer vertical resolution within the PBL. Identical
vertical layer distributions are maintained across all three horizontal resolutions (25 km,
5 km, and 1 km), ensuring direct comparability of turbulent mixing across different
horizontal resolutions. A total of 50 terrain-following vertical eta-layers extending from
the surface to approximately 15 km were used in all three resolution simulations, with
30 layers distributed below 2 km above the ground to describe the atmospheric
boundary structure in detail. The vertical layer was strategically designed with 7 layers

below 200 meters (each approximately 20 meters in height), 3 layers between 200 and 300 meters (each about 30 meters in height), and 8 layers between 300 and 1000 meters (each approximately 80 meters in height). This configuration comprehensively captures mixed layer development and key turbulent processes (e.g., entrainment and surface flux exchange) through layer densification, which is sufficient to capture PBL turbulent mixing. Jiang et al. (2024) and Jiang and Hu (2023) have demonstrated that the number of model vertical layers primarily influences vertical distribution, with more vertical grid layers producing a more stable vertical structure under stable boundary conditions that better resolves boundary layer turbulence.

In order to allow for a straightforward comparison of multi-resolutions simulations and facilitate the identification of differences between the high- and low-resolution simulations, the corner locations of the 1 km and 5 km resolution domains are aligned with the corner locations of the 25 km grid cell. Each grid cell in the 25 km simulation consists of a 5 x 5 set of cells from the 5 km simulation, and each grid cell in the 5 km simulation comprises 5 x 5 cells from the 1 km simulation, as shown in Figure S3. Thus, exactly 25 grids at 5 km resolution and 625 grids at 1 km resolution are embedded within each 25 km grid cell.

To ensure similar boundary forcing across the three simulations, initial and boundary conditions are handled differently for the 25 km, 5 km, and 1 km resolution domains. For the 25 km resolution, meteorological initial and lateral boundary conditions are obtained from the National Center for Environmental Prediction (NCEP) final reanalysis (FNL) data with $1° \times 1°$ resolution and 6 h temporal resolution. Initial and boundary conditions for the trace gases and aerosol species are provided by the quasi-global WRF-Chem simulation with 360 x 145 grid cells (67.5°S-77.5°N, 180°W-180°E) at $1° \times 1°$ resolution. The initial and boundary conditions for the simulation at 5 km resolution are derived from the simulation at 25 km resolution. Similarly, the initial and boundary conditions for the simulation at 1 km resolution are derived from the simulation at 5 km resolution. In this way, since the forcing for the study area is consistent across multi-resolutions, differences in simulation results among multi-resolutions can be attributed to disparities in model resolutions.


2.1.3 Emissions

Anthropogenic emissions for the outer quasi-global simulation are derived from the

Hemispheric Transport of Air Pollution version-2 (HTAPv2) at 0.1° x 0.1° horizontal
resolution and a monthly temporal resolution for 2010 (Janssens-Maenhout et al., 2015).
The Multi-resolution Emission Inventory for China (MEIC) at 0.25° x 0.25° horizontal
resolution for 2019 (Li et al., 2017a; Li et al., 2017b) is used to replace emissions over
China within the simulation domain. Emission differences significantly contribute to
pollutant concentration variability across multi-resolutions. Qian et al. (2010) showed
that sub-grid variability of emissions can contribute up to 50% of the variability near
Mexico City. To eliminate the impact of inconsistent emissions on pollutant
concentrations simulated at multi-resolutions, we ensured emission consistency across
all three domains by interpolating emissions for all species from the 25 km resolution
domain to both the 5 km and 1 km resolution domains. This study primarily focuses on
BC. The spatial distribution of BC emissions is shown in Figure 2. Figure S4 illustrates
BC emissions at three different resolutions, demonstrating similar spatial patterns
across multi-resolutions. Biomass burning emissions are obtained from the Fire
Inventory from NCAR (FINN) at 1 km horizontal resolution and an hour temporal
resolution (Wiedinmyer et al., 2011). The diurnal variation of biomass burning
emissions follows the suggestions by WRAP (2005), with injection heights based on
Dentener et al. (2006) from the Aerosol Comparison between Observations and Models
(AeroCom) project. Biogenic emissions were calculated using the Model of Emissions
of Gases and Aerosols from Nature (MEGAN) v3.0 model (Guenther, 2007; Zhang et
al., 2021).

**2.2 Observational data**
2.2.1 Meteorological data

The meteorological data were obtained from the observation tower at the University

of Science and Technology of China (USTC) in Hefei, Anhui, China (117.27°E,
31.84°N), indicated by a solid black triangle in Figure 1a. The tower measures

temperature, relative humidity, wind speed, and wind direction at 2 m, 4.5 m, 8 m, 12.5

m and 18 m heights. This site represents a typical urban surface within the study area.

The tower was installed on the roof of a teaching building, with its top 17 m above the

canopy plane. It is equipped with three RM Young 03002 anemometers and three

HPM155A temperature and humidity sensors to measure the aforementioned

meteorological parameters (Yuan et al., 2016; Liu et al., 2017). This study focuses on

analyzing temperature, relative humidity, and wind speed.

Additionally, we employed meteorological data from automatic weather stations

(AWSs), which were established based on the operational standards issued by the China

Meteorological Administration (CMA, 2018). The hourly data underwent quality

control (QC) by local meteorological bureaus of Anhui, following World

Meteorological Organization guidelines (Estevez et al., 2011). The QC included checks

of consistency, such as internal, temporal-spatial, and climatic range validations. These

QC data were used to determine daily mean, minimum, and maximum meteorological

variables. The AWSs recorded various parameters, including air temperature (T, °C),

wind speed (U, m/s), air pressure (P, Pa), and wind direction. In this study, we focus on

the 3-hourly 2 m temperature and 10 m wind speed obtained from four AWS stations

located in the study region. The four AWS sites are marked by purple solid dots in

Figure S5.

2.2.2 Pollutants data

We used the hourly BC observations from the air quality monitoring site on the

campus of USTC during spring (March 10 to March 20, 2019). In this study, we focus

on analyzing BC observational data to compare with model output. BC was observed

using a Multi-angle Absorption Photometer (MAAP, Model-5012) manufactured by

Thermo Scientific. This instrument is located approximately 260 m north of the USTC

meteorological tower. It takes advantage of the strong visible light absorption properties

of BC aerosols. There is a linear relationship between the attenuation of the beam after

passing through the aerosol sample and the load of BC aerosols on the fiber membrane.

The BC concentration is derived by inverting this relationship. A light scattering

measurement is incorporated into the chamber to correct for multiple scattering effects caused by particle accumulation on the filter tape. The MAAP-5012 Black Carbon Meter collects atmospheric aerosols using glass fiber filter membranes and observes them at a wavelength of 670 nm.

Although this study primarily focuses on the simulation of BC, we conducted a comprehensive validation of other air pollutants to ensure the reliability of the simulation results. However, after being initially obtained via a parameterized PBL scheme, the mixing coefficients for gases are then clipped to empirically chosen thresholds of 1 $m^2$/s over rural regions and 2 $m^2$/s over urban regions, with the distinction between rural and urban regions made based on the local CO emission strength. Thus, the boundary layer mixing coefficient for gases in the WRF-Chem model is implicitly influenced by emission resolution rather than directly controlled by model resolution. Consequently, the existing adjustment process for gas mixing coefficients, which relies on CO emission strength, is unsuitable for studying the impact of model resolution on the turbulent mixing of gaseous pollutants. In contrast, the mixing coefficients for particulate matter are directly calculated through boundary layer parameterization without subsequent modifications. The publicly available version of WRF-Chem defines a default lower limit of 0.1 m²/s for particulate matter mixing coefficients. We did not implement the adjustment proposed by Du et al. (2020), which suggest raising the lower limit of PBL mixing coefficient from 0.1 $m^2$/s to 5 $m^2$/s within the PBL. Although setting specific thresholds can improve simulation results, such thresholds are predominantly empirical in nature, whether based on CO and $PM_{2.5}$ emissions or the 5 m²/s threshold suggested by Du et al. (2020). These threshold adjustments effectively compensate for missing physical processes in the model by artificially enhancing mixing intensity. Our approach focuses on understanding the physical mechanisms responsible for the model's underestimation of nighttime mixing intensity, with particular emphasis on how model resolution affects turbulent mixing processes. Rather than employing empirical thresholds to align model output with observations, we aim to investigate the fundamental causes of the discrepancies. We contend that threshold approaches rely heavily on empirical data, lack sufficient

theoretical foundation, and may impede comprehensive understanding of the underlying physical processes. Consequently, this study utilizes the default particulate matter turbulent mixing coefficients in the model for our analyses. In this study, we limited our additional validation to $PM_{2.5}$ (fine particulate matter with aerodynamic diameters less than 2.5 µm), as its mixing processes are governed by the same resolution-dependent mechanisms as BC. Ground observations of hourly $PM_{2.5}$ surface concentrations during March 2019 were obtained from the website of the Ministry of Environmental Protection of China (MEP of China). As our study concentrates on the Hefei region, we selected 10 monitoring stations within this area for detailed analysis. These stations are marked as black triangles in Figure S5.

While hourly observations for both meteorology and pollutants are available, model outputs are provided at 3-hour intervals to balance computational efficiency and storage requirements. Hourly output data would provide higher time resolution but significantly increase storage demands. Given that we ran simulations at multi-resolutions (25 km, 5 km, and 1 km), hourly outputs would have generated prohibitively large data volumes. On the other hand, this 3-hour output interval remains sufficient for our primary research objective of analyzing daily pollutant variations (particularly BC) rather than precise hourly comparisons. This approach effectively captures daily variability patterns without losing essential detail. For direct comparisons, hourly observations were sampled to match our 3-hour model output intervals.

## 3. Results

### 3.1 Simulated meteorological fields at various horizontal resolutions

Meteorological fields may play a crucial role in the turbulent mixing and pollutant transport. In this study, we evaluate time series of simulated temperature, wind speed, and relative humidity across three resolutions against observational data to assess resolution impacts on these key meteorological variables. Figure 3a compares the time series of observed and simulated 8-m wind speeds at the USTC site (117.27°E, 31.84°N). Simulation results among multi-resolutions are similar, attributing to

relatively flat and uncomplicated topography. The temporal trends of the simulations closely align with observational data, exhibiting distinct diurnal variations characterized by higher values during the daytime and lower values at night. Additionally, the model struggles to capture some moments accurately, overestimating wind speed when it suddenly increases. For instance, on March 20 at noon, while the observed peak wind speed is approximately 6 m/s, simulations at 25 km and 5 km resolutions produced maximum wind speeds of approximately 9 m/s, significantly exceeding the observed value, with only the 1 km resolution simulation yielding results close to the observation. Figure 3b compares the 2-m temperature simulated at three different resolutions with the observation. The multi-resolutions simulation results exhibit remarkable consistency and closely align with observations. Temperature displays a pronounced diurnal variation, fluctuating between 5 and 30 ℃ with relative stability. However, the model occasionally underestimates or overestimates values at certain time points. As shown in Figure 3c, the multi-resolutions simulated results demonstrate consistency and accurately capture the diurnal variation trend of observed relative humidity (RH). Model results are highly consistent with observations, both reaching a maximum of 100%.

Additionally, Figure S6 displays the time series of observed and simulated meteorological variables averaged across four AWS stations in the study region. Figure S6a presents a comparison of 10-m wind speed simulated at three different resolutions, revealing generally consistent results with observations. The overall pattern is similar to that observed at the single USTC station, characterized by a clear diurnal variation with higher wind speeds during daytime and lower speeds at night. However, simulations at all three resolutions occasionally deviate from observations. For example, on March 11, the 5 km and 1 km resolution models overestimate wind speed at approximately 7 m/s compared to the observed 4 m/s. Conversely, on March 14 during the daytime, all three resolutions underestimate wind speed, simulating around 2 m/s against an observed value of 4 m/s. Figure S6b compares the simulated 2-m temperatures across three resolutions with observational data. The simulated temperatures are remarkably similar across all resolutions and show strong correlation

with observations throughout most of the study period. Only a few outliers were noted,
which minimally impact the overall pattern. For example, all resolution models
overestimate temperature at noon on March 20, simulating approximately 28°C while
the observed temperature is only about 20°C.
In summary, the simulated meteorological variables across multi-resolutions
demonstrate strong similarity and closely match the observations, with only occasional
minor discrepancies. However, our subsequent analysis reveals that the variations in
pollutant concentrations across multi-resolutions cannot be attributed to the minor
discrepancies observed in the time series of meteorological variables.

**3.2 Simulated BC surface concentrations and impacts of turbulent mixing at**
**various horizontal resolutions**
3.2.1 Surface concentrations simulated at three different horizontal resolutions
The spatial distribution of BC surface concentrations across multi-resolutions in the
study area is illustrated in Figure 4. As the resolution improves from 25 km to 5 km and
further to 1 km, BC surface concentrations reveal more detailed spatial features. Figure
4a presents the simulation results across multi-resolutions, averaged over the whole day.
Significant variations exist from coarse resolutions to fine resolutions, with surface
concentrations decreasing as resolution increases from 25 km to 5 km and further to 1
km. BC surface concentrations range from 0 to 9 ug/m$^3$. At 25 km resolution, there is a
notable discrepancy between the spatial distributions of BC concentrations and
emissions (Fig. 2). The highest simulated concentration at 25 km resolution is located
west of the USTC site, while maximum emissions are centered at the USTC site. Our
analysis indicates that the difference in turbulent mixing between these two regions
leads to spatial inconsistency between BC surface concentrations and emissions. The
details of this phenomenon will be discussed in section 3.2.2. Figure 4b illustrates the
spatial distribution of BC surface concentrations during the daytime. The differences in
surface concentrations among multi-resolutions are minimal, with values falling within
the range of 0 to 5 ug/m$^3$. In the central urban areas, the BC surface concentration
simulated at 25 km resolution is marginally lower than those simulated at finer
resolutions. Moreover, during the daytime, simulated BC concentrations over Chaohu
lake areas are notably higher than in other regions, potentially due to the impact of dry
deposition velocity. Figure S7 shows the spatial distribution of dry deposition velocity,
revealing lower values over lakes compared to other areas. This lower dry deposition
velocity leads to higher pollutant concentrations over lakes compared to land areas after
pollutants transport to the lake surface during the daytime. At night, dry deposition
velocity is similar to that of surrounding non-urban land areas. Consequently, nighttime
BC concentrations over lakes are approximately equal to those in surrounding areas.
Figure 4c demonstrates the spatial distribution of BC surface concentrations during
nighttime. Compared to daytime, BC surface concentrations are notably higher in all
major urban regions at night, with high-resolution simulations capturing more spatial
variations. In conclusion, BC surface concentrations decrease as resolution increases
from 25 km to 5 km and further 1 km. However, the spatial distribution of BC surface
concentrations at 5 km and 1 km resolutions are similar throughout the whole day.
To facilitate a more accurate and direct comparison of results across multi-
resolutions, we refine coarse grids to match fine grids. The detailed refinement process
is described in Text S1. Figure S8a exhibits the spatial differences in BC surface
concentrations between 25 km and 5 km resolutions, as well as between 25 km and 1
km resolutions, averaged over the whole day. It reveals that coarse-resolution (25 km
resolution) simulations generally yield higher BC surface concentrations than fine-
resolutions (5 km and 1 km resolution) simulations across most areas. The largest
disparities mainly occur in central urban areas with complex underlying surfaces and
complicated flow patterns. Figure S8b demonstrates the spatial differences in BC
surface concentrations between 25 km and 5 km resolutions, as well as between 25 km
and 1 km resolutions during the daytime, revealing smaller disparities mostly ranging
between -1 and 1 ug/m$^3$. In contrast, Figure S8c depicts pronounced differences in BC
concentrations between 25 km and 5 km resolutions, as well as between 25 km and 1
km resolutions during the nighttime, with most areas exhibiting disparities exceeding 2
ug/m$^3$. The largest differences are mainly concentrated in urban areas. These findings
indicate that diversities in BC surface concentrations among multi-resolutions are
primarily attributable to nocturnal concentrations in urban areas. However, differences
between 5 km and 1 km resolutions are small compared to those between 25 km and
finer resolutions (5 km and 1 km). BC surface concentrations are approximately equal
in the 5 km and 1 km simulations, as shown in Figure S9.

Furthermore, BC observations from the USTC monitoring station were utilized to

validate the simulated BC surface concentrations. Figure 5 illustrates the diurnal
variation of BC surface concentrations averaged over the USTC site. Both observations
and simulations exhibit a pronounced diurnal variation, with lower concentrations
during the daytime and higher concentrations at night. During the daytime, BC surface
concentrations simulated at three resolutions are comparable to the observational data.
However, nighttime simulations significantly overestimate BC surface concentrations.
As resolution increases from 25 km to 5 km and 1 km, the simulated surface
concentrations decrease, aligning more closely with observations. The 25 km resolution
simulations yield the highest concentrations, with a maximum value of approximately
12 ug/m$^3$, nearly double the observed values. In contrast, BC surface concentrations
simulated at 5 km and 1 km resolutions are similar and more closely align with
nocturnal observations, peaking at around 9 ug/m$^3$. In conclusion, the diurnal variation
of the observation is better captured by high-resolution (5 km and 1 km) simulations.
The performance of BC surface concentrations across multi-resolutions demonstrates
that coarse grid spacing inadequately captures local pollutant distributions.

To verify the accuracy and comprehensiveness of the simulation results, we further

analyzed the diurnal variation of $PM_{2.5}$ surface concentrations. Figure S10 illustrates
the diurnal variation of simulated $PM_{2.5}$ surface concentrations across multi-resolutions
compared with observations. The diurnal pattern of $PM_{2.5}$ closely resembles that of BC,
characterized by higher concentrations at night and lower concentrations during
daytime. Across all resolutions, the model slightly underestimates daytime $PM_{2.5}$
surface observations while overestimating nighttime values. Notably, increased
horizontal resolution substantially improves nocturnal simulations. The 25 km
resolution simulation generates an anomalous midnight peak (105 μg/m³), resulting in
a +61% bias, whereas the 5 km and 1 km resolutions substantially mitigate these

deviations to approximately 30%. To further examine the contribution of each $PM_{2.5}$ component to the diurnal variation across multi-resolutions, Figure S11 shows the diurnal variations of four $PM_{2.5}$ constituents (sulfate ($SO_4^{2-}$), nitrate ($NO_3^-$), OIN, and organic carbon (OC)) averaged over 10 MEP sites in Hefei. Significant differences emerge in the diurnal variations of these components across multi-resolution simulations. Specifically, the surface concentrations of $NO_3^-$, OIN, and OC exhibit a consistent diurnal pattern, with lower concentrations during daytime and higher concentrations at night. As resolution increases from 25 km to 5 km and 1 km, the simulated components surface concentrations decrease, aligning more closely with observations.

The total concentration of $PM_{2.5}$ and its components demonstrates significant sensitivity to horizontal resolutions. Coarse resolution simulations underestimate turbulent mixing capacity, resulting in overestimated concentrations. Higher resolution simulations more accurately capture vertical mixing within the PBL. For secondary particles such as sulfates and nitrates, formation rates depend heavily on local precursor substance concentrations ($SO_2$, $NO_x$). Higher resolution simulations may enable more realistic representation of precursor substance diffusion, leading to reduced local concentration gradients and consequently slower secondary aerosol formation rates. Additionally, variations in $PM_{2.5}$ surface concentrations across multi-resolutions may also stem from complex secondary particle generation mechanisms. For instance, liquid-phase oxidation of sulfates in clouds is sensitive to local cloud water distribution, with higher resolutions better capturing small-scale cloud structures that potentially alter sulfate formation efficiency. The formation of ammonium nitrate ($NH_4NO_3$) is particularly sensitive to temperature and humidity variations. At higher resolutions, temperature and humidity gradients induced by urban heat island effects or topographical variations can be more realistically simulated, influencing the distribution of gaseous nitric acid ($HNO_3$) and particulate nitrate ($NO_3^-$). Dry deposition processes may also contribute to resolution-dependent variations, as local differences in surface roughness (including buildings and vegetation) become more apparent at higher resolutions, directly affecting particulate deposition velocity rates. Overall, the

simulation results for major air pollutants fall within a reasonable error range compared
to observational data, confirming the reliability of the model for this study.

We now aim to further investigate the underlying factors contributing to the

discrepancies in atmospheric pollutant simulations, with a particular focus on BC,
across different spatial resolutions. Previous studies have indicated that the diurnal
variation of atmospheric particulate matter concentrations is primarily controlled by
daily variations of PBL mixing and pollutants emissions (Du et al., 2020). The diurnal
variation of BC emissions peak during the daytime and are lower at night. During
nighttime, pollutants are trapped within the shallow boundary layer due to the reduced
turbulent mixing, resulting in elevated surface concentrations of atmospheric
particulate matter. As the boundary layer develops in the morning, pollutants rapidly
diffuse and are transported to upper atmospheric layers, leading to relatively low
surface concentrations. Therefore, the turbulent mixing process plays a crucial role in
determining pollutant concentrations.

3.2.2 Impacts of turbulent mixing on BC surface concentrations at three different
horizontal resolutions

To investigate the vertical mixing depth influencing pollutant diffusion, we first

analyze the PBL height, as illustrated in Figure 6. Figure 6a shows the spatial
distribution of the PBL height simulated at three different resolutions, averaged over
the whole day. Higher-resolution simulations yield lower PBL heights and capture more
intricate details compared to lower-resolution simulations. This trend is consistent
during both daytime and nighttime. Figure 6b demonstrates that the PBL height exceeds
0.9 km across most regions during the daytime. Notably, due to strong topographic
influences, the PBL height in the vicinity of Chaohu Lake is remarkably low, typically
less than 0.1 km. Conversely, in the southwestern region, characterized by higher
elevations and more complex terrain, the PBL height surpasses 1.1 km. Figure 6c
depicts the nighttime PBL heights at three different resolutions. These heights
predominantly fall below 0.3 km, significantly lower than those during the daytime.
The PBL height gradually decreases as the resolution increases, which should typically

lead to higher BC surface concentrations. However, BC surface concentrations actually decrease as resolution increases from 25 km to 5 km and 1km (Figure 4). Consequently, the PBL height alone cannot explain the differences in pollutant simulations among multi-resolutions in this study.

Previous studies have established that PBL mixing coefficients are critical determinants in air quality modeling (Du et al., 2020). In WRF-Chem, turbulent mixing within the boundary layer is partially governed by PBL mixing coefficients generated by the PBL parameterization scheme. It is worth noting that the mixing coefficients for atmospheric particulate matter and gases are two distinct variables in the current version of WRF-Chem. The boundary layer mixing coefficient for gases is initially obtained via a parameterized PBL scheme but undergoes additional modification through an empirical parameterization that enhances gas mixing based on CO emission strength (Kuhn et al., 2024). This enhancement applies to gas pollutants when using the MOSAIC aerosol scheme, as implemented in this study. Specifically, gas mixing coefficients are clipped to empirically chosen thresholds of 1 $m^2$/s over rural regions and 2 $m^2$/s over urban regions, with the distinction between rural and urban regions made based on the local CO emission strength. In contrast, the mixing coefficient of particulate matter is directly calculated through boundary layer parameterization without subsequent modifications. Our study focuses exclusively on the turbulent mixing of atmospheric particulate matter, analyzing the aerosol mixing coefficient with the default lower limit of 0.1 $m^2$/s as specified in the publicly released version of WRF-Chem. We have not implemented the mixing coefficient adjustments proposed by Du et al. (2020), which suggest raising the lower limit of PBL mixing coefficient from 0.1 $m^2$/s to 5 $m^2$/s within the PBL. We contend that threshold approaches are primarily based on empirical data and may impede comprehensive understanding of the underlying physical processes. In our study, particulate matter mixing coefficients are directly calculated through boundary layer parameterization without adjustments based on empirical settings. This approach allows the model to more accurately represent the natural turbulent mixing processes. Consequently, we can investigate the turbulent mixing intensity of particulate matter across different horizontal resolutions and

examine the true impact of grid resolution on pollutant mixing.

The spatial distribution of aerosol turbulent mixing coefficients at the lowest model

layer is analyzed, as shown in Figure 7. Figure 7a illustrates the simulation results
across multi-resolutions averaged over the whole day. The variations in PBL mixing
coefficients across different resolutions are evident, with high-resolution simulations
capturing more spatial characteristics. The spatial distribution of the PBL mixing
coefficient demonstrates strong correlation with land use type and terrain height, which
will be explored subsequently. Turbulent mixing coefficients range from 0 to 8 $m^2/s$,
with peak values predominantly located in urban areas. Notably, the mixing coefficient
simulated at 25 km resolution near surface around USTC substantially exceeds that of
the western area, resulting in lower BC surface concentrations simulated at 25 km
resolution at USTC compared to its western regions (Figure 4). This discrepancy leads
to a mismatch between the spatial distribution of pollutant concentrations and emissions,
as discussed in section 3.2.1. During the daytime, the PBL mixing coefficients
simulated at three resolutions are relatively high, ranging from 0 to 17 $m^2/s$, as shown
in Figure 7b. BC masses simulated across multi-resolutions are fully mixed within the
boundary layer, resulting in similar BC surface concentrations across these resolutions.
Conversely, turbulent mixing coefficients diminish considerably during the nighttime,
with maximum values approximately 3 $m^2/s$, as shown in Figure 7c. The turbulent
mixing coefficient emerges as one of the important factors controlling surface pollutant
concentrations under stable nocturnal PBL conditions. Nighttime PBL coefficients are
higher at 5 km and 1 km resolutions compared to 25 km resolution across most of the
study area, resulting in lower BC surface concentrations at these two higher resolutions
during the nighttime. Figure S12 further illustrates the disparities in parametrized PBL
mixing coefficients between 25 km resolution and the two higher-resolution simulations.
However, Figure S13 shows that the turbulent mixing coefficient parameterized at 5 km
resolution is larger than that at 1 km resolution, which fails to explain the similar surface
concentrations in these two higher-resolution (5 km and 1 km) simulations. To further
investigate this phenomenon, we selected a meridional section passing through the
USTC site to analyze the distribution of vertical wind speed flux, which represents the
turbulent mixing directly resolved by large-scale dynamic processes.

Figure 8 displays the cross section of meridional wind speed flux along the USTC

site simulated at three different resolutions. The upward vertical wind speed flux
simulated at 25 km resolution are near the surface. However, the 5 km resolution
simulation generates stronger upward motion at a slightly higher altitude, specifically
between 850 and 1000 hPa. Notably, the 1 km resolution simulation captures the highest
vertical wind speed flux, with relatively intensive upward motion extending beyond
500 hPa. The 1 km resolution can resolve small-scale eddies and capture the most
pronounced vertical wind speed fluxes. In comparison, simulations at 5 km resolution
are able to capture smaller-scale eddies, while those at 25 km resolution occasionally
capture larger-scale eddies. Despite the larger PBL mixing coefficients at 5 km
resolution compared to 1 km resolution near the surface, the upward vertical wind speed
flux at 1 km resolution reaches higher altitudes, indicating the presence of more small-
scale eddies and resulting in enhanced vertical turbulent mixing. Consequently, near the
surface, the combined effects of turbulent mixing, which is represented by both the
parameterized PBL mixing coefficient and the directly resolved vertical wind speed
flux, lead to similar BC surface concentrations at higher resolutions (5 km and 1 km)
simulations. Furthermore, Figure S14 shows the meridional cross section during
daytime and nighttime. During the day, the mixing height is relatively high at all three
resolutions, allowing pollutants to be fully mixed and transported within the PBL. This
results in similar BC surface concentrations across multi-resolutions. Conversely, at
night, high-resolution simulations resolve more small-scale eddies, resulting in vertical
transport reaching higher altitudes and intensifying turbulent mixing. In conclusion,
pollutants in lower-resolutions (25 km) simulations tend to accumulate near the surface,
whereas at higher resolutions (5 km and 1 km) simulations, pollutants are transported
to higher heights. This phenomenon contributes to imparities in BC surface
concentration across multi-resolutions.

3.2.3 Impacts of land use type and terrain height on turbulent mixing coefficients at
three different horizontal resolutions
Previous analysis indicate that the PBL mixing coefficient is one of the main factors
contributing to the disparities in BC surface concentrations across multi-resolutions.
Therefore, we further explored the factors influencing the spatial distribution of the
PBL mixing coefficient. Our analysis reveals that the spatial distribution of the PBL
mixing coefficient is closely related to land use types and terrain height. Specifically,
the overall distribution of the turbulent mixing coefficient is closely resembled by the
land use types (Figure 1b and Figure 7). However, in areas with obvious magnitude
changes, such as east of the USTC site, the turbulent mixing coefficient displays distinct
gradient changes that are not reflected in land use patterns. Notably, the spatial
distribution of the topographic height (Figure 1a) in this region exhibits distinct gradient
changes similar to those of the turbulent mixing coefficients. Consequently, the spatial
distribution of the turbulent mixing coefficient is influenced by both terrain and land
use types. This correlation can be attributed to the inter-relationship among turbulent
mixing, friction velocity, terrain, and land use types. Terrain and land use types
influence friction velocity by modifying surface roughness, which in turn directly
affects turbulent mixing coefficients within the PBL. Higher surface roughness
typically lead to greater fiction velocity, subsequently enhancing turbulent intensity and
increasing the vertical mixing efficiency of pollutants within the PBL. To further
investigate this relationship, the spatial distribution of friction velocity is analyzed, as
shown in Figure 9. The analysis reveals that friction velocity increases as resolution
increases from 25 km to 5 km and 1 km resolutions, with finer resolutions (5 km and 1
km) capturing more spatial details. Differences in friction velocity are illustrated in
Figure S15. The spatial distribution of friction velocity indeed correlates with terrain
and land use patterns, consequently influencing the distribution of the PBL mixing
coefficient. As a result, the spatial distribution of the PBL mixing coefficient correlates
with land use types and terrain height.
Our study indicates that variations in land use type distribution simulated at
different horizontal resolutions are a significant factor causing changes in PBL mixing
coefficients across multi-resolutions. These variations in mixing coefficients relate
closely to BC surface concentrations, explaining specific patterns of BC surface
concentration distributions. For example, the BC surface concentration south of the
USTC site increases as resolution improves from 25 km to 5 km and 1 km resolutions
(Figure 4 and Figure S8), contrasting with concentration variations simulated in other
regions. Our analysis reveals that the turbulent mixing coefficient simulated at 25 km
resolution is higher compared to the two higher-resolution simulations in this area
(Figure 7 and Figure S12). Moreover, the spatial distribution of land use types indicates
that the 25 km resolution simulation resolves only a single urban land use type in this
area (Figure 1b). In contrast, higher resolution simulations capture additional land use
types beyond the urban, including lakes, farmland, and shrubs (Figure 1b). The
inclusion of these diverse land use types in the higher resolution leads to smaller PBL
mixing coefficients in this area, as the surface roughness associated with lakes,
farmland, and shrubs is generally lower than that of urban areas. As a result, the reduced
vertical mixing in the finer resolution (5 km and 1 km) simulations results in higher BC
surface concentrations south of the USTC site.
Additionally, to explore the differences in PBL mixing coefficients across multi-
resolutions under uniform land use conditions, we designed another set of sensitivity
experiments across three resolutions. As mentioned earlier, the only difference from the
baseline experiment was the use of the default USGS land use classification data in the
WRF model. As shown in Figure S2, land use type data at different horizontal
resolutions are approximately consistent in this setup. All other settings remained
identical to those in the baseline experiment.
Figure 10 presents the spatial distribution of PBL mixing coefficients in the
sensitivity experiment. Figure 10a illustrates the results across multi-resolutions
averaged over the whole day. Similar to the baseline experiment, increasing resolution
resolves more spatial details. For example, in the area where the USTC site is located,
the PBL mixing coefficient in the 25 km resolution simulation of the sensitivity
experiment is approximately 4.3 m²/s, significantly lower than the 8 m²/s observed in
the baseline experiment. This pattern is consistent across higher resolutions (5 km and
1 km). This finding aligns with the spatial distribution of land use types used in both
sets of experiments (Figure 1b and Figure S2). The decrease in mixing coefficients in
the sensitivity experiment stems from its land use data failing to resolve urban land
types in urban areas. Figures 10b and 10c show the PBL mixing coefficients of the
sensitivity experiment during daytime and nighttime, respectively. Consistent with the
baseline experiment, the turbulent mixing coefficients during the day are substantially
higher than at night. The PBL coefficients in the nighttime simulations are higher at 5
km and 1 km resolutions compared to the 25 km resolution.

Additionally, Figure S16 further illustrates the differences in the parameterized

PBL mixing coefficient between the 25 km resolution and the two higher-resolution
simulations under roughly uniform land use conditions. Figure S16c shows that in the
city center, the boundary layer mixing coefficient parameterized at 5 km and 1 km
resolutions is higher than at the 25 km resolution during nighttime. Since urban areas
are primarily flat, topographical differences between different resolutions in urban areas
are minimal, almost negligible. Furthermore, because the land use types in the
sensitivity experiment are approximately consistent across different resolutions, the
main factor responsible for resolution-related differences in the PBL mixing
coefficients in urban areas is the grid size. Notably, in areas with significant topographic
variations, such as suburban and rural regions, the difference in boundary layer mixing
coefficients between 25 km and 5 km/1 km resolutions in the sensitivity experiment
strongly correlates with the spatial distribution of topographic differences. This directly
demonstrates that topographic height is also a key determinant of boundary layer
mixing coefficient distribution. Qian et al. (2010) indicated that the terrain affects the
transport and mixing of aerosols and trace gases, as well as their concentrations across
multi-resolutions, through its impact on meteorological fields such as wind and the PBL
structure. These terrain-related effects are particularly significant in regions with more
variable topography. Additionally, Figure S17 shows that the turbulent mixing intensity
parameterized at 5 km resolution in the sensitivity experiment is greater than at 1 km
resolution. Further analysis of the latitude-pressure cross section of BC concentrations
and vertical wind speed flux, as shown in Figure S18, indicates that, similar to the
baseline experiment, the 1 km resolution of the sensitivity experiment resolves more
small-scale turbulent eddies, capturing more prominent vertical wind speed flux, thus
resulting in stronger turbulent mixing.

Through comprehensive analysis of both baseline and sensitivity experiments, we

found that within the resolution range of 25 km to 5 km and 1 km, the spatial distribution
accuracy of land use types plays a decisive role in parameterizing the PBL mixing
coefficient. Finer land use type information at higher resolutions directly alters the
spatial distribution of the boundary layer mixing coefficient, with urban surfaces
significantly increasing the parameterized PBL mixing coefficient. Therefore,
accurately representing land use types, particularly urban surfaces, is critical for
parameterizing the PBL mixing coefficient. On the other hand, in the sensitivity
experiment, complex terrain areas with significant elevation (such as suburban, rural,
and hilly regions) increase mixing coefficients by enhancing surface roughness,
whereas this effect is weaker in flat urban areas. Consequently, differences in PBL
mixing coefficients across multi-resolutions strongly correlate with terrain precision.
Higher resolutions can resolve finer terrain variations, affecting local turbulent mixing
(such as terrain-induced mechanical turbulence). This confirms the dominant role of
high-resolution terrain and land use information in PBL mixing coefficient
parameterization. Notably, in regions where land use types and terrain height remain
relatively flat and consistent across different horizontal resolutions in the sensitivity
experiments, increasing resolution still leads to enhanced boundary layer mixing
coefficients, highlighting the importance of grid size for parameterizing the boundary
layer mixing coefficient. In the resolution range from 5 km to 1 km, higher resolution
slightly reduces the parameterized boundary layer mixing coefficient. However, the 1
km resolution model resolves more small-scale turbulent eddies, resulting in stronger
turbulent mixing at night. In summary, for parameterization of boundary layer mixing
coefficients across multi-resolutions, high-resolution surface information is more
important in regions with significant changes in land use types and terrain height. Grid
size is also crucial in regions with more gradual changes, where higher-resolution grids
consistently enhance boundary layer mixing representation. Therefore, to improve PBL
mixing coefficient simulation, priority should be given to ensuring accuracy of land use
data (especially spatial representation of urban types), precise terrain representation in

complex regions, and appropriated grid resolution to enhance turbulent mixing simulation.

**3.3 Simulated BC column concentrations and impacts of turbulent mixing at various horizontal resolutions**

3.3.1 Simulated BC column concentrations at three different horizontal resolutions

It is generally accepted that the turbulent mixing process primarily affects pollutant surface concentrations by mixing surface pollutants into higher layers, without altering the column concentration. However, in this study, BC column concentrations exhibit differences across multi-resolutions simulations. Therefore, we further investigate the spatial distribution of BC column concentrations and the main mechanisms behind these variations. Figure 11a illustrates the spatial distribution of BC column concentrations simulated at three resolutions, averaged over the whole day. The regional average values for the three resolutions are 2041, 2150, and 2223 ug/m$^2$, respectively. The 5 km and 1 km resolution simulations yield larger BC column concentrations compared to 25 km resolution simulations. The spatial distribution of BC column concentrations simulated at 25 km resolution is highly consistent with the BC emission distributions (Figure 2), showing high concentrations in central urban areas exceeding 2500 ug/m$^2$, while regions distant from urban centers demonstrate lower concentrations, generally below 2100 ug/m$^2$. The 5 km resolution simulation results indicate peak column concentrations concentrated in urban areas and spread around, with the southwestern area approaching 2250 ug/m$^2$. The 1 km resolution simulation results yield the largest BC column concentrations and demonstrate the most pronounced diffusion tendency, with most areas exceeding 2250 ug/m$^2$. Figure 11b and Figure 11c reveal lower BC column concentrations during the daytime compared to those at night, with a more pronounced dispersion trend of column concentrations simulated at night. Figure S19 depicts the differences in BC column concentrations between 25 km and 5 km resolutions, as well as between 25 km and 1 km resolutions, revealing that BC column concentrations in coarser resolutions are marginally lower than those in finer resolutions (5 km and 1 km) in most of the study areas. On the other

hand, the BC column concentration simulated at 1 km resolution are larger than those
at 5 km resolution, as shown in Figure S20. In conclusion, BC column concentrations
increases with increased resolutions, accompanied by a more pronounced dispersion
tendency towards higher and farther areas.

3.3.2 Impacts of turbulent mixing on BC column concentrations at three different
horizontal resolutions

We further analyze the mechanisms underlying the differences in BC column

concentrations across multi-resolutions in urban areas. Figure 12a displays the vertical
profiles of BC concentrations averaged over the study area. The BC profiles at 25 km
resolution exhibit significant variability, generally decreasing from the surface to higher
altitudes. The near-surface BC concentration is approximately three times higher than
those at high altitudes, with surface concentrations reaching about 3 ug/m$^3$. At an
altitude of 100 m, the concentration drops to 1 ug/m$^3$, while above this elevation, the
BC concentration is less than 1 ug/m$^3$. Substantial disparities exist among multi-
resolutions simulations in the vertical profiles of BC concentrations. Our analyses
above have shown that near the surface, the parameterized mixing coefficients and
directly resolved vertical wind speed flux are lower at 25 km resolution compared to 5
km and 1 km resolutions, reducing the vertical mixing of pollutants in 25 km resolution
simulations. Thus, BC concentrations at 25 km resolution are higher near the surface
and lower at higher altitudes compared to high-resolution (5 km and 1 km) simulations.
Moreover, the parametrized PBL mixing coefficient at 1 km resolution is lower than at
5 km resolution in the atmosphere, but the directly resolved upward vertical wind speed
flux by the model dynamic process reaches higher altitudes at 1 km resolution compared
to 5 km resolution. Thus, due to the combined effects of these two processes, the
intensity of turbulent mixing is similar between the 5 km and 1 km resolutions at near-
surface levels, whereas it is greater at 1 km resolution than at 5 km resolution at higher
altitudes. In numerical models, sub-grid scale (SGS) turbulent diffusion is typically
simulated by parameterization schemes. However, as model resolution increases, such
as achieving 1 km resolution, the turbulent mixing is increasingly resolved by
dynamical framework of the model. This advancement allows the model to capture
dynamic structures and small-scale turbulence more accurately, significantly enhancing
the strength of turbulent mixing. The resolution of dynamic processes reduces reliance
on traditional parameterization schemes, thereby decreasing the PBL mixing coefficient
parameterized at finer resolutions. In conclusion, at higher altitudes, the enhanced
turbulent mixing efficiently facilitates more ground-emitted pollutants to higher height
as resolution increases. Thus, BC concentrations at 5 km and 1 km resolution are similar
near surface, with 1 km resolution yielding the largest concentrations at higher altitudes.

To further investigate the BC column concentrations and their dispersion tendency

towards farther areas, we analyzed the vertical profile of wind speed at three resolutions
averaged over the study area, as shown in Figure 12b. The vertical profile of wind speed
is relatively consistent across the three resolutions. From the ground to higher altitudes,
the overall wind speed gradually increases, transitioning from low speeds near the
surface to higher speeds aloft. Near the ground, the simulated average wind speed is
approximately 1 m/s, increasing to 4 m/s at an altitude of 1 km, and reaching an average
of about 7 m/s at an altitude of 2 km. In the upper atmosphere, characterized by larger
wind speeds, pollutants mixed up from near-surface can be transported and dispersed
farther. As previously mentioned, BC simulated in higher-resolution simulations can be
transported to higher altitudes, thus dispersing over greater distances by stronger winds.
Therefore, as the resolution increases, the trend of diffusion towards farther regions in
the simulated BC column concentrations becomes more pronounced.
As previously discussed, higher-resolution simulations facilitate BC transport to
greater altitudes and further distances. This phenomenon extends its atmospheric
lifetime, consequently resulting in increased column concentrations. Bauer et al. (2013)
noted that turbulent mixing and convective transport processes play a critical role in
determining BC lifetimes. Figure 13 illustrates the spatial distribution of BC lifetime,
calculated by dividing the BC column concentration by the dry deposition flux. It
demonstrates that BC lifetime gradually lengthens as resolution increases. The average
lifetime of BC column concentrations in the study area is 344 h, 350 h, and 382 h for
25 km, 5 km, and 1 km resolutions, respectively. These results clearly demonstrate that
BC simulated at higher resolutions exhibits prolonged atmospheric residence times.
Consequently, the BC column concentration is higher in high-resolution simulations.

## 4. Conclusion and Discussion

Turbulent mixing plays a crucial role in urban pollutant transport by enhancing the
diffusion of atmospheric pollutants. Current atmospheric models often underestimate
turbulent exchange within stable nocturnal boundary layers, and the turbulent mixing
varies markedly across different model horizontal resolutions. However, few studies
have analyzed how turbulent mixing processes across multi-resolutions affect pollutant
concentrations in urban areas. Therefore, our goal is to elucidate the variations in
pollutant concentrations across multi-resolutions and investigate the influence of
turbulent mixing on pollutant concentrations at various resolutions.
We conducted a three-nested WRF-Chem simulation at 25 km, 5 km, and 1 km
resolutions in the Hefei area. BC surface concentrations decrease as resolution increases
from 25 km to 5 km and further to 1 km but are similar at 5 km and 1 km resolutions,
showing significant diurnal variations with higher concentrations at night and lower
during the daytime. The BC surface concentrations across multi-resolutions align well
with USTC site observations during daytime but are overestimated at night, with this
overestimation decreasing at higher-resolution (5 km and 1 km). Disparities in BC
surface concentrations between the two finer-resolution and the 25 km resolution
simulations are primarily attributable to nocturnal concentrations. In addition, the
diurnal variation of $PM_{2.5}$ surface concentrations simulated at different resolutions
follows the same trend as the observed concentrations at the national monitoring sites,
with slight underestimation during daytime and overestimation at night. The PBL
mixing coefficient plays a crucial role in controlling surface particulate matter
concentrations at night. Larger nighttime PBL mixing coefficients and higher vertical
wind speed flux at 5 km and 1 km resolutions compared to 25 km resolution near the
surface result in lower BC surface concentrations. However, the PBL mixing coefficient
at 5 km is larger than at 1 km resolution. Moreover, the upward vertical wind speed flux

resolved at 1 km resolution reaches higher altitudes compared to 25 km and 5 km resolutions, indicating more small-scale eddies and resulting in enhanced turbulent mixing. Consequently, near the surface, the combined effects of parametrized PBL mixing coefficient and the directly resolved vertical wind speed flux lead to similar BC surface concentrations at 5 km and 1 km resolutions.

Further analysis reveals that the spatial distribution of PBL mixing coefficients is influenced by both land use types and terrain heights. The turbulent mixing coefficient correlates with the spatial distribution of land use types at smaller scales, with urban underlying surfaces notably increasing the parameterized PBL mixing coefficient. The mixing coefficient also strongly correlates with terrain heights at larger scales, particularly in regions with complex topography and significant elevation differences, where higher terrain substantially enhances mixing coefficients. This correlation can be attributed to the interrelationship among turbulent mixing coefficients, friction velocity, terrain, and land use types. The static database of terrain and land use types employed as model input determines the surface roughness. Higher surface roughness typically leads to greater fiction velocity, subsequently increasing the PBL mixing coefficients. Moreover, in regions where land use types and terrain height remain relatively flat and consistent across multi-resolutions, increasing resolution still enhances boundary layer mixing coefficients, highlighting the importance of grid size. Thus, both surface information and grid resolution are crucial for accurately parameterizing PBL mixing coefficients, with priority given to accurate land use data, precise terrain representation, and higher grid resolution to improve turbulent mixing simulations.

In WRF-Chem, the mixing coefficients of chemical species are clipped to empirically chosen thresholds of 1 $m^2$/s over rural areas and 2 $m^2$/s over urban areas to prevent unrealistically low values. These thresholds are modified based on differences in anthropogenic CO and primary $PM_{2.5}$ emissions between rural and urban regions. Importantly, this adjustment applies exclusively to gases and not to aerosols when the MOSAIC or MADE/SORGAM aerosol schemes are used. This is because the adjustment does not couple with the aerosol PBL mixing scheme in WRF-Chem, although potential modifications could be made for compatibility with the MOSAIC or

MADE/SORGAM scheme. Thus, the boundary layer mixing coefficient for gases is
implicitly influenced by emission resolution rather than directly controlled by model
resolution. In this study, this treatment caused gas mixing coefficients to converge
across different horizontal resolutions, preventing us from accurately assessing the
impact of horizontal resolution on gas turbulent mixing. For aerosols, however, the
original PBL mixing coefficients are retained, which are directly parametrized from
boundary layer parameterization schemes. Therefore, our focus is mainly on particulate
matter in this analysis and we omitted this modification for gases.
The variations in turbulent mixing across multi-resolution simulations not only
affect the BC surface concentration but also lead to different BC column concentrations.
BC column concentrations increase with improved resolutions, accompanied by a more
pronounced diffusion tendency towards higher altitudes and distant regions.
Throughout the atmosphere, turbulent mixing intensifies with improved resolutions,
resulting in pollutants being transported to higher altitudes. Concurrently, wind speed
increases with altitude, facilitating the pollutants which are mixed to higher altitudes to
be spread farther. Consequently, BC simulated at higher resolution is transported to
greater altitudes and dispersed to farther regions, thus persisting in the atmosphere for
longer periods and leading to larger lifetimes. As a result, BC column concentrations
increase with finer resolutions.
This study highlights the importance of model horizontal resolution in simulating
the dispersion of atmospheric pollutants. We observed that the enhanced turbulent
mixing strength in high-resolution can more accurately reproduce the vertical and
horizontal distribution of pollutants, thus aligning the simulated pollutant surface
concentrations more closely with actual observations. In contrast, turbulent mixing in
low-resolution simulations, primarily depending on boundary layer parameterizations,
may not adequately capture the dynamics of turbulence, leading to discrepancies
between the simulated and actual distribution of pollutants, particularly during the night
with stable boundary condition. Future research should focus on improving PBL
parameterization schemes to enhance model performance at lower resolutions, thereby
better serving the needs of air pollution control and environmental management.
We have noted that the parameterized PBL mixing coefficient decreases when
transitioning from 5 km to 1 km resolution, alongside an increase in vertical wind speed
flux which represents turbulent mixing directly resolved by the dynamical processes.
This trend suggests that if the resolution was further increased to LES scales, the
parameterized PBL mixing coefficient might diminish significantly, potentially
approaching zero, while the turbulence mixing resolved directly by the dynamics would
intensify considerably. At LES scales, the majority of turbulent mixing is directly
resolved, capturing the atmospheric dynamical processes and turbulent exchanges more
realistically, thereby reducing the simulation biases caused by parameterization errors.
This shift diminishes reliance on traditional boundary layer parameterizations to
simulate turbulent mixing, leading to a substantial reduction in the parameterized
boundary layer mixing coefficient. By capturing the finer details of atmospheric
dynamics, the model provides a more realistic representation of turbulent mixing and
related physical processes, which is crucial for understanding weather patterns, climate
variability, and pollutant dispersion. However, due to the huge computational resources
required for LES simulation, we have not yet performed an analysis at the LES scale,
but it is worth further exploring in the future.
Moreover, in addition to the influence of surface roughness on turbulence intensity,
surface type significantly affects the CBL and turbulence mixing strength through
differences in radiative flux absorption, reflection, and heat exchange. There are
substantial variations in the absorption and reflection of shortwave radiation across
different surface types. Urban areas typically have lower albedo, absorbing more
shortwave radiation, which increases surface temperature and transfers energy to the
atmosphere as sensible heat. In contrast, vegetated areas generally have higher albedo
and, through transpiration, release more latent heat while reducing sensible heat output.
These differences in energy exchange between surface and atmosphere directly
influence turbulence strength. Furthermore, the varying balance between sensible and
latent heat fluxes across different surface types impacts turbulence intensity and CBL
depth. For instance, urban areas, with stronger sensible heat flux, tend to generate more
intense thermal convection, often resulting in a shallower CBL, while vegetated areas,
with predominant latent heat flux, may develop more stable atmospheric conditions,
potentially leading to a deeper CBL with weaker turbulence. These mechanisms of
radiative absorption and heat exchange are crucial in the formation of the diurnal CBL
and determining turbulence intensity. Future studies on land use impacts on turbulence
mixing should therefore consider not only surface roughness but also radiative flux
differences, sensible and latent heat exchange mechanisms, and the comprehensive
effects of surface albedo on turbulence development.
Our analysis also found that higher-resolution facilitate transport over greater
distances, suggesting that inter-city pollutant diffusion can be affected by model
resolution, with coarse-resolution potentially reducing long-range transport and inter-
urban impacts. While previous studies have examined pollutant formation mechanisms
at specific resolutions and explored the physical and chemical interactions among
megacities, few have considered the impacts of different resolutions on long-range
transport between cities. Due to computational cost constraints, inter-urban impacts are
not discussed in this study but deserve further investigation in the future. Finally, while
vertical resolution is held constant in our study, we recognize that it could influence the
interpretation of the turbulence processes in certain scenarios, especially in regions with
complex vertical structures. Therefore, future work could systematically explore the
interplay between vertical resolution and pollutant concentration or aerosol-boundary
layer feedbacks.

*Data availability.* The release version of WRF-Chem can be downloaded from http://www2.mmm.ucar.edu/wrf/users/download/get_source.html. The updated USTC version of WRF-Chem can be downloaded from http://aemol.ustc.edu.cn/product/list/ or contact chunzhao@ustc.edu.cn. Additionally, code modifications will be incorporated into the release version of WRF-Chem in the future.

*Author contributions.* ZY and CZ designed the experiments and conducted and analyzed the simulations. All authors contributed to the discussion and final version of the paper.

*Competing interests.* The contact author has declared that none of the authors has any competing interests.

*Acknowledgments.* This research was supported by the National Key Research and Development Program of China (No. 2022YFC3700701), the Strategic Priority Research Program of Chinese Academy of Sciences (XDB0500303), National Natural Science Foundation of China (41775146), the USTC Research Funds of the Double First-Class Initiative (YD2080002007, KY2080000114), the Science and Technology Innovation Project of Laoshan Laboratory (LSKJ202300305), and the National Key Scientific and Technological Infrastructure project "Earth System Numerical Simulation Facility" (EarthLab). The study used the computing resources from the Supercomputing Center of the University of Science and Technology of China (USTC) and the Qingdao Supercomputing and Big Data Center.

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

**Table 1** WRF-Chem model configuration

| Horizontal resolution | 25 km & 5 km & 1 km |
|---|---|
| Domain size | 140 x 105 & 250 x 250 & 150 x 150 |
| Simulation period | 5 March to 21 March 2019 |
| Gas-phase chemistry scheme | SAPRC99 mechanism |
| Radiation scheme | Fast-J |
| PBL scheme | YSU scheme |
| Microphysics scheme | Morrison two-moment scheme |
| Land surface scheme | Noah land-surface scheme |
| Cumulus scheme | Kain-Fritsch (25 km grid only) |
| Surface layer scheme | Revised MM5 Monin-Obukhov scheme |
| Longwave radiation scheme | RRTMG scheme |
| Shortwave radiation scheme | RRTMG scheme |


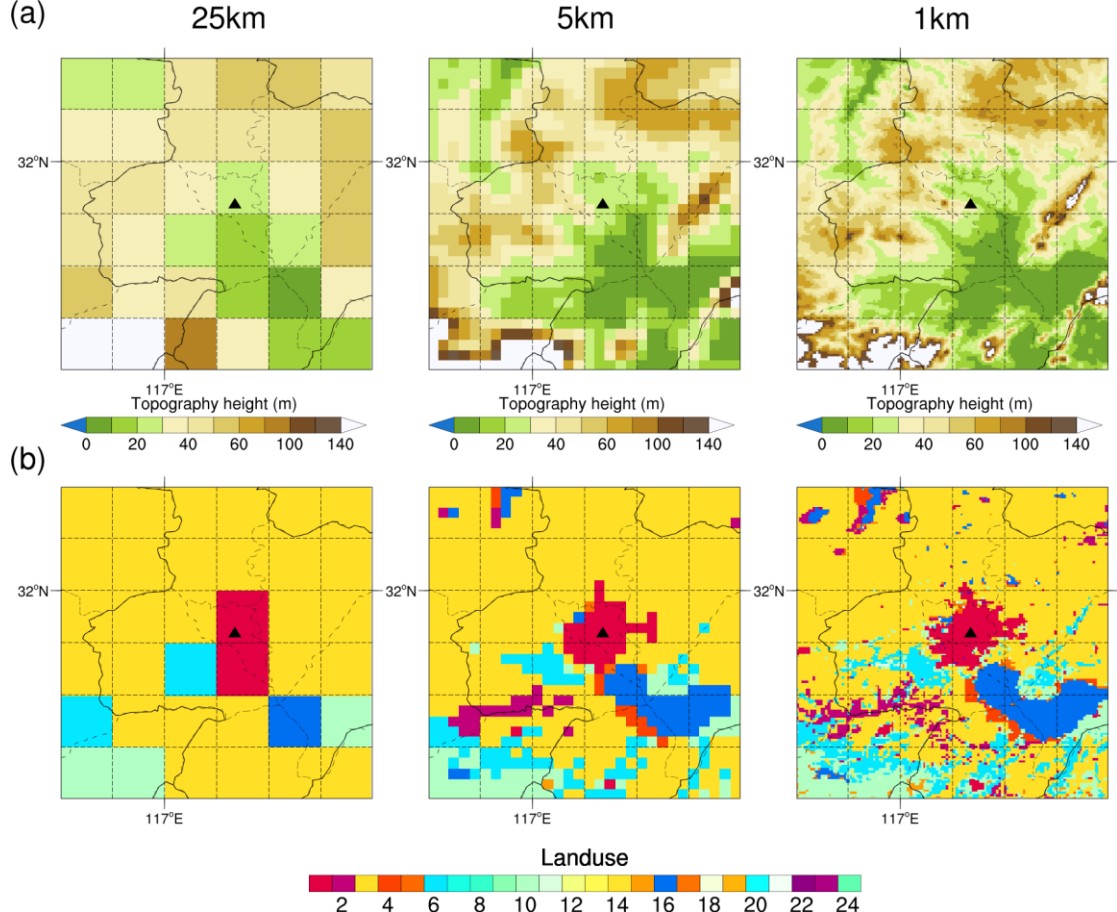

**Figure 1.** (a) The terrain height (m) in the study area for 25-km (left), 5-km (middle), and 1-km (right) resolution simulations, respectively; (b) Spatial distribution of land use types in the study area for 25-km (left), 5-km (middle), and 1-km (right) resolution simulations, respectively. The solid black triangle indicates the location of the USTC site.

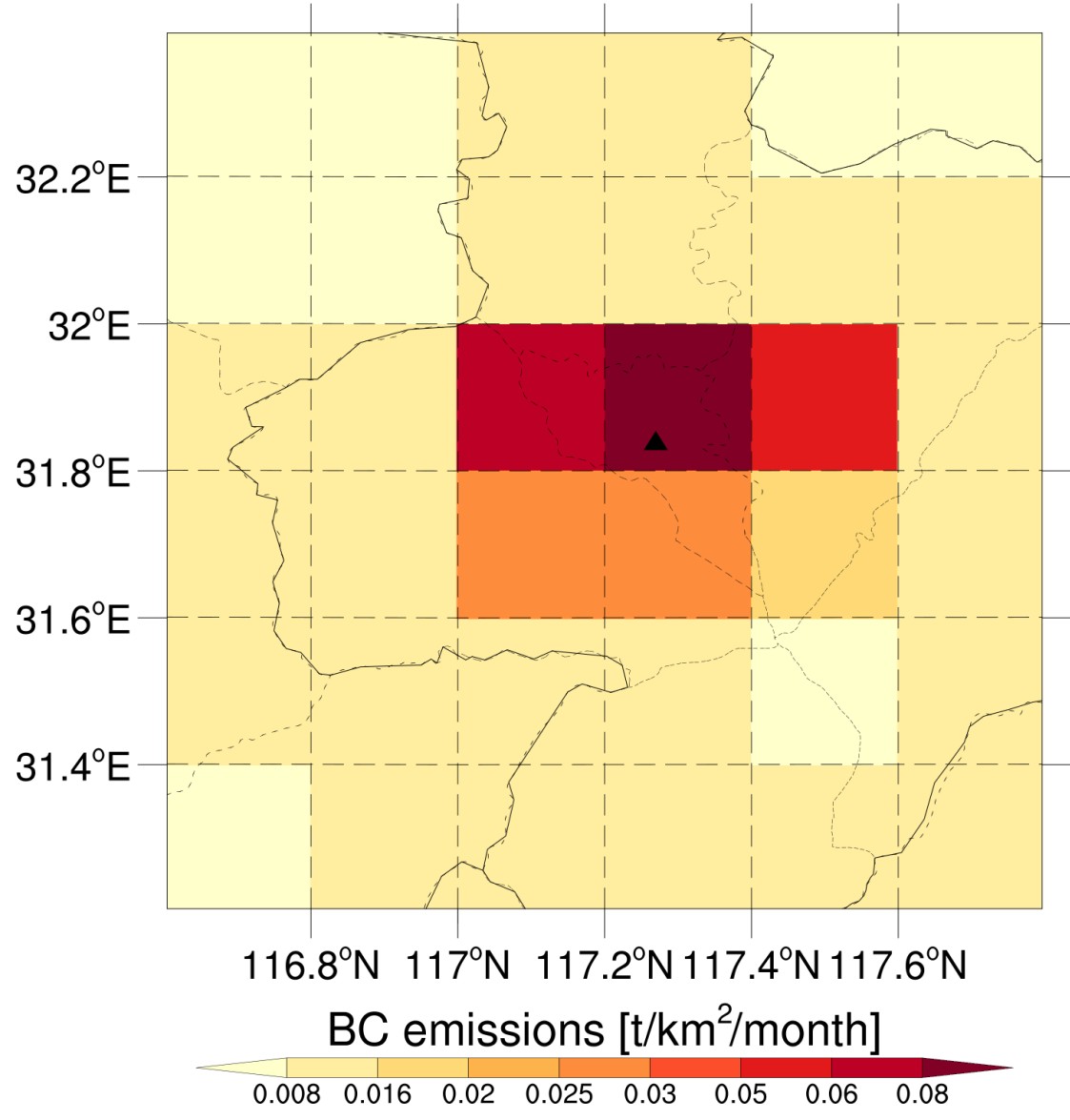

**Figure 2.** Spatial distribution of BC emissions in the study area. The solid black triangle indicates the location of the USTC site.

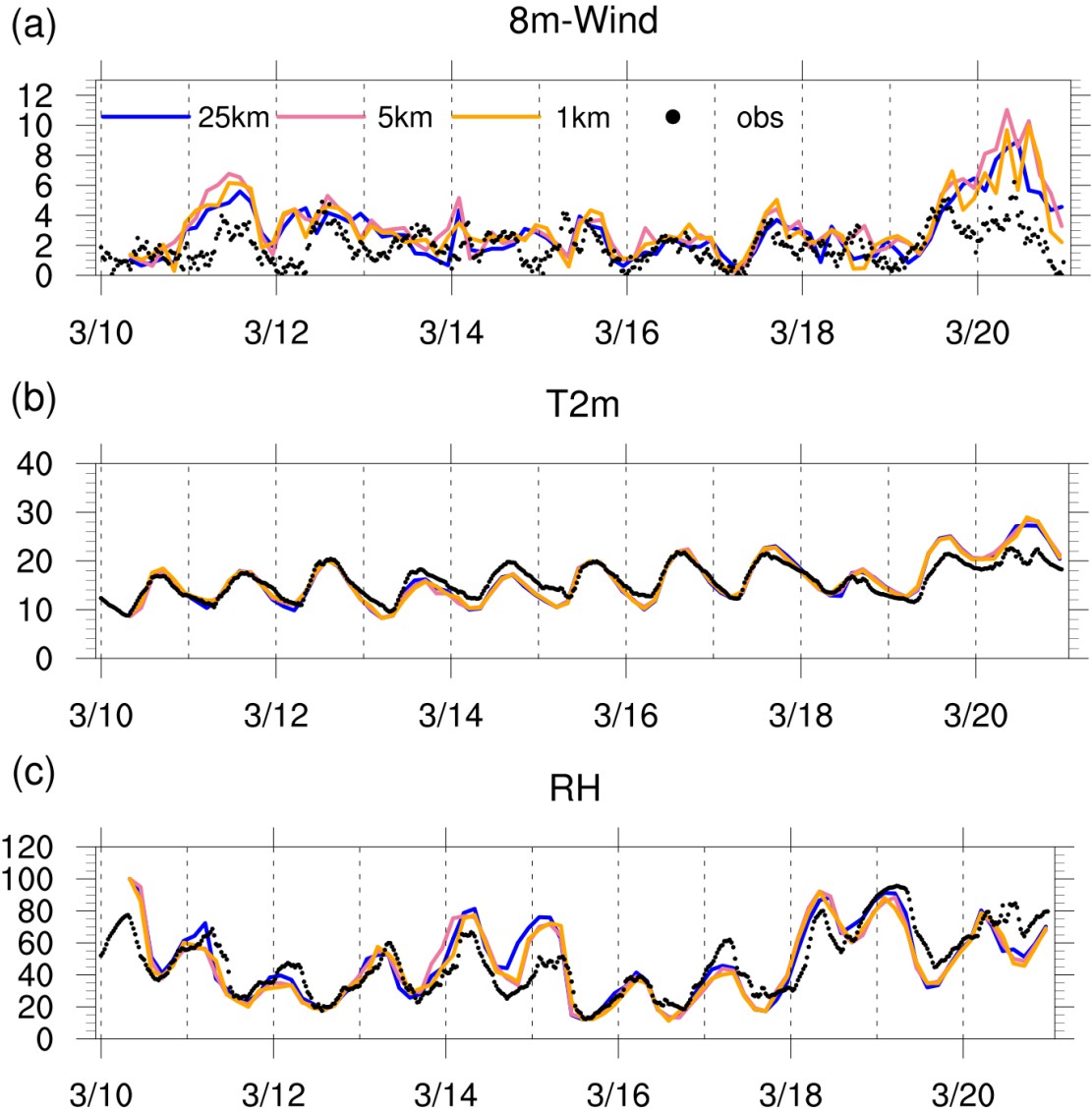

**Figure 3.** Time series at USTC meteorological tower observation site of observed

(black dot) and simulated wind speed at 8 m (top, unit: m s$^{-1}$), temperature at 2 m

(middle, unit: °C), and relative humidity (bottom, unit: %) for 25-km (solid blue line)

resolution, 5-km (solid pink line) resolution, and 1-km (solid orange line) resolution,

respectively.

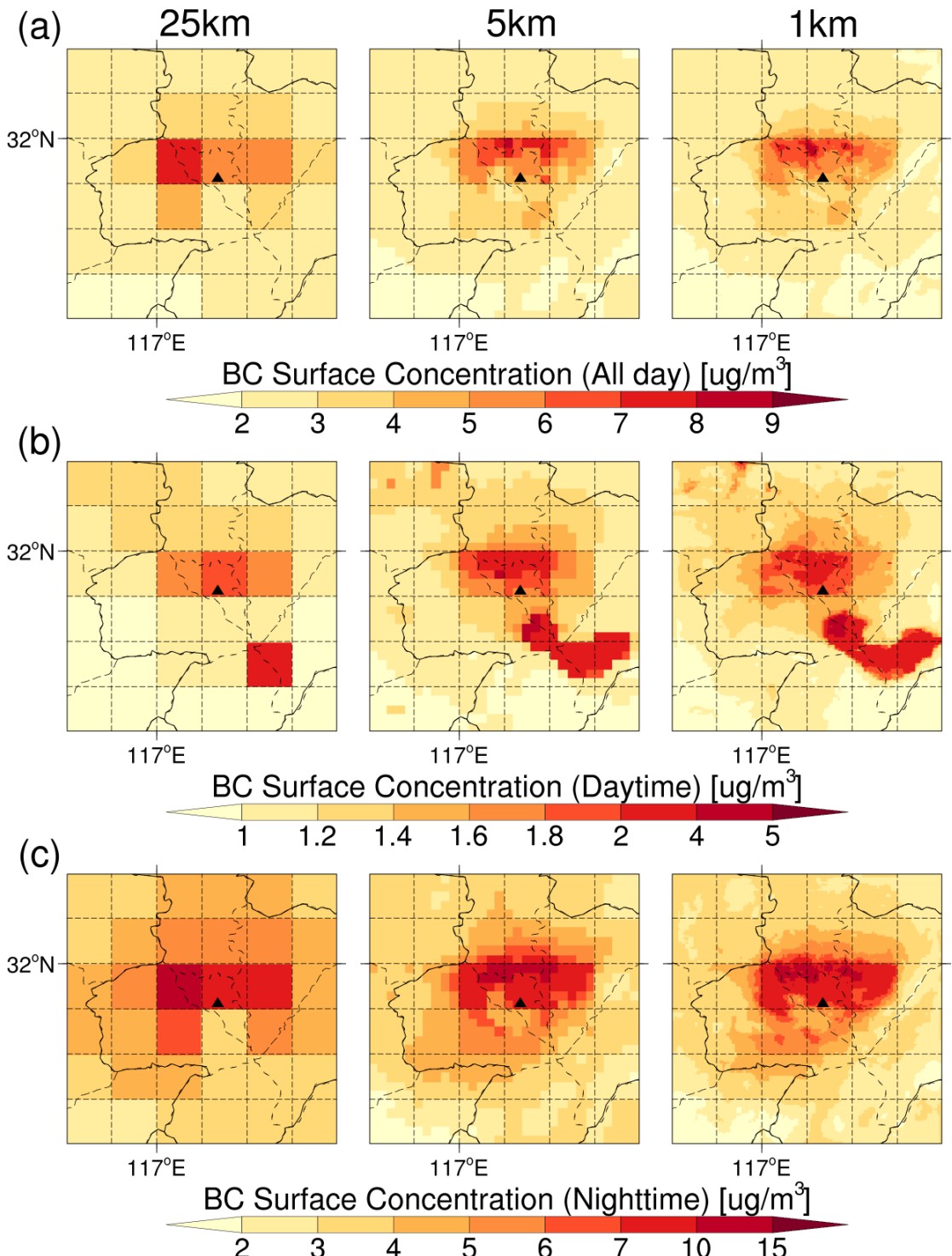

**Figure 4.** Spatial distribution of the BC surface concentration in the study area for 25-km (left), 5-km (middle), and 1-km (right) resolution simulations of the whole day (top), the daytime (middle), and the nighttime (bottom), respectively. The solid black triangle indicates the location of the USTC site.

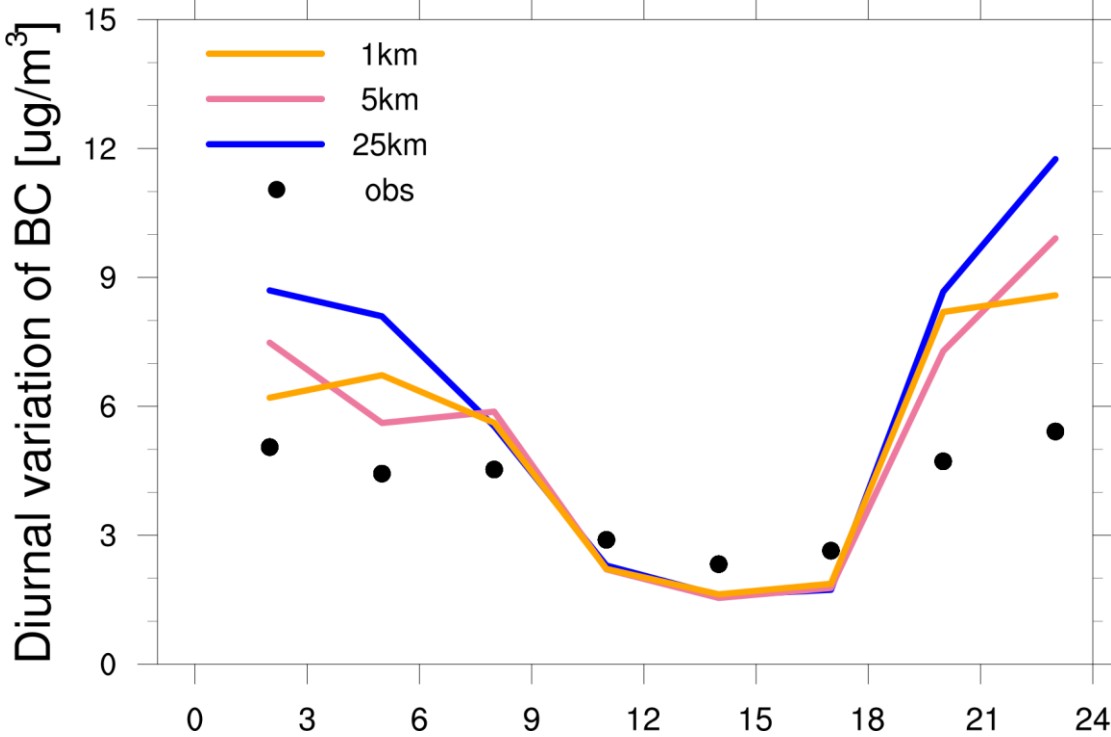

**Figure 5.** Diurnal variation of BC surface concentrations within 24 h averaged over
the USTC site during the study period for 25-km (solid blue line), 5-km (solid pink
line), and 1-km (solid orange line) resolution simulations and observations (black
dot). Both the simulated results and observations are sampled at the model output
frequency, i.e., 3-hourly.

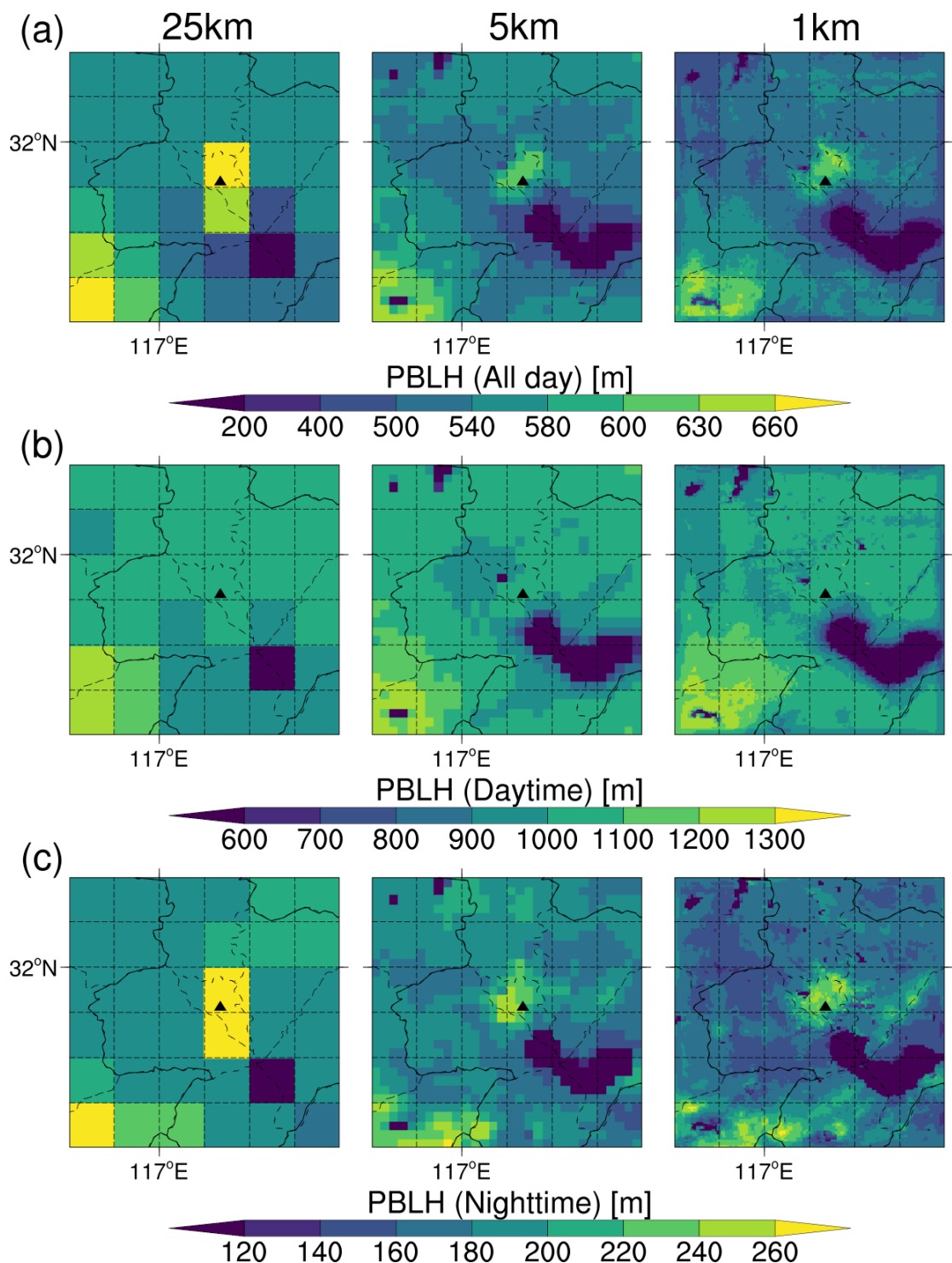

**Figure 6.** Spatial distribution of the PBL height in the study area for 25-km (left), 5-km (middle), and 1-km (right) resolution simulations of the whole day (top), the daytime (middle), and the nighttime (bottom), respectively. The solid black triangle indicates the location of the USTC site.

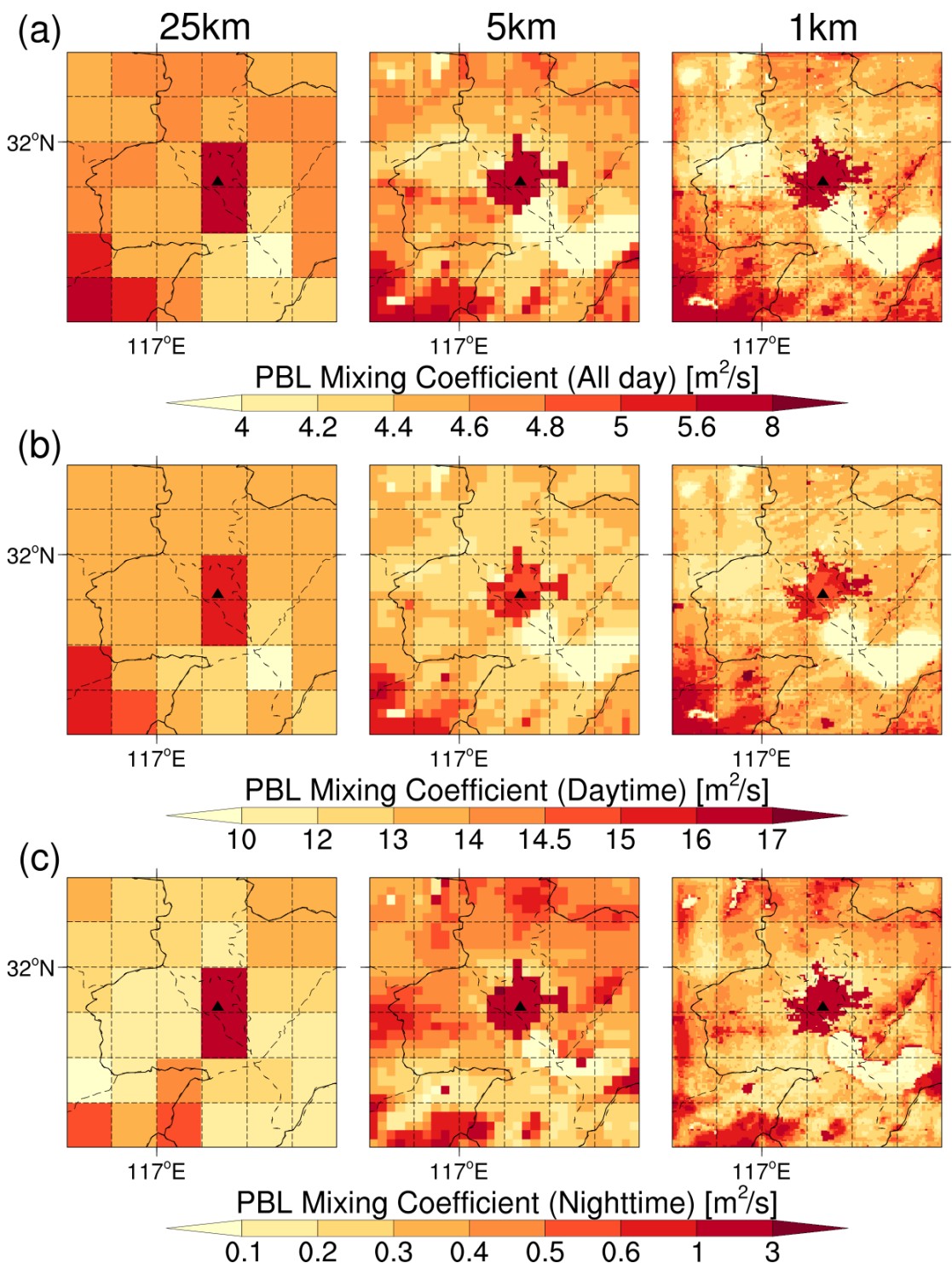

1565

**Figure 7.** Spatial distribution of PBL mixing coefficients in the study area for 25-km

(left), 5-km (middle), and 1-km (right) resolution simulations of the whole day (top),

the daytime (middle), and the nighttime (bottom), respectively. The solid black

triangle indicates the location of the USTC site.

1570

1571

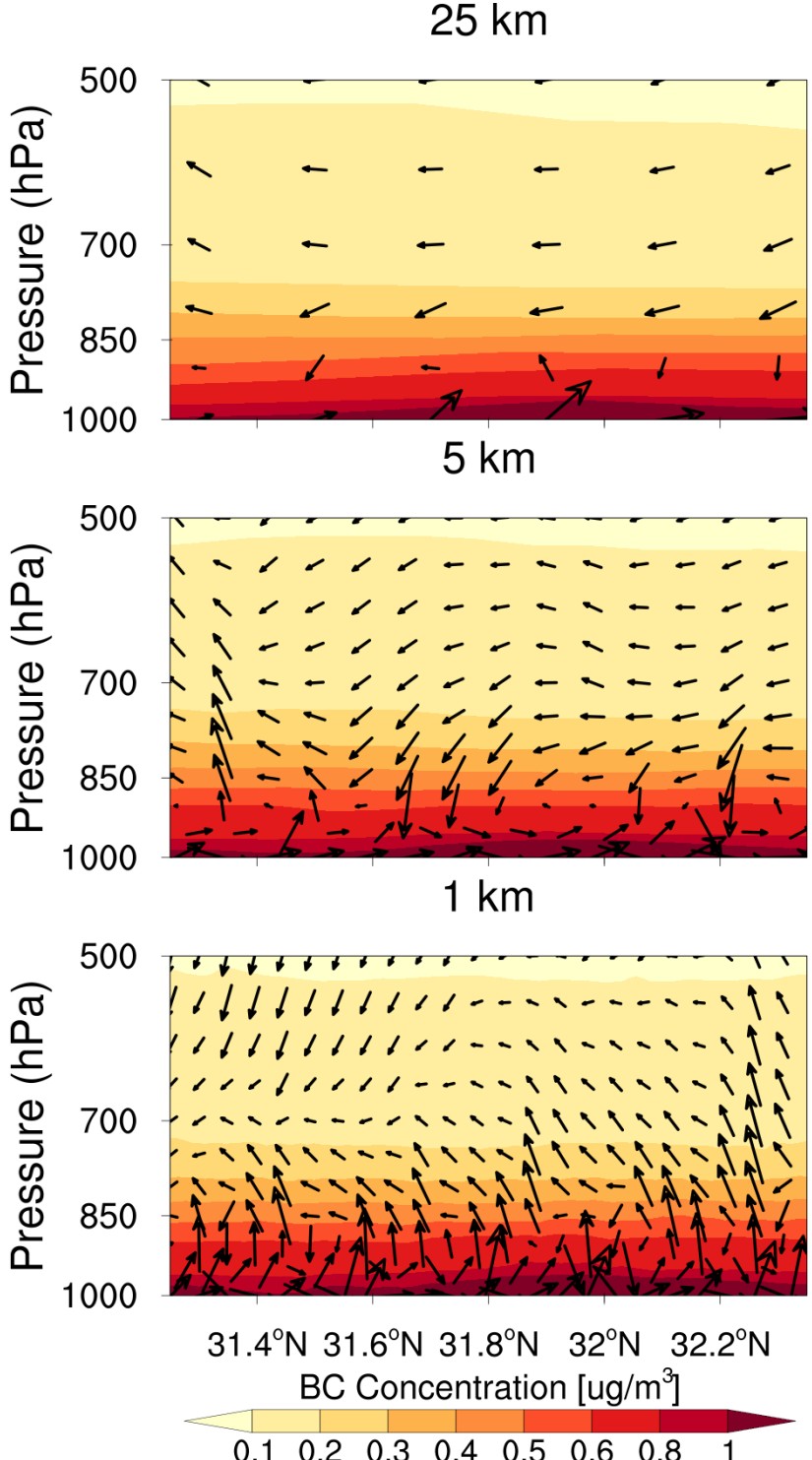

**Figure 8.** The latitude-pressure cross section of BC concentrations and wind speed flux along the USTC site for 25-km (top), 5-km (middle), and 1-km (bottom) resolution simulations of the whole day, respectively. Vector arrows are the combination of wind speed fluxes v and w, with the vertical wind speed flux being multiplied by 100 for visibility. The shaded contours represent BC concentrations at each pressure level.

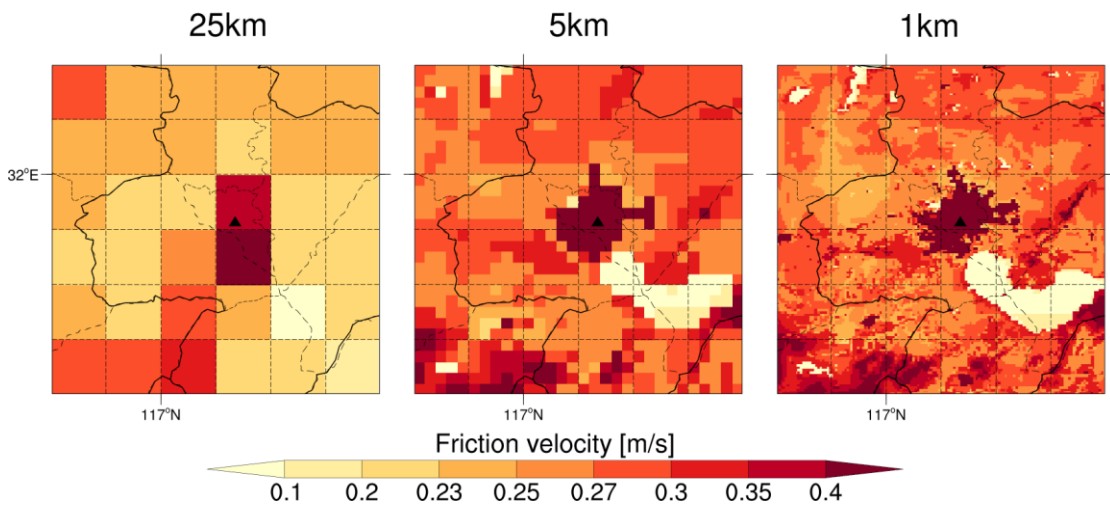

**Figure 9.** Spatial distribution of friction velocity in the study area for 25-km (left), 5-km (middle), and 1-km (right) resolution simulations of the whole day, respectively. The solid black triangle indicates the location of the USTC site.

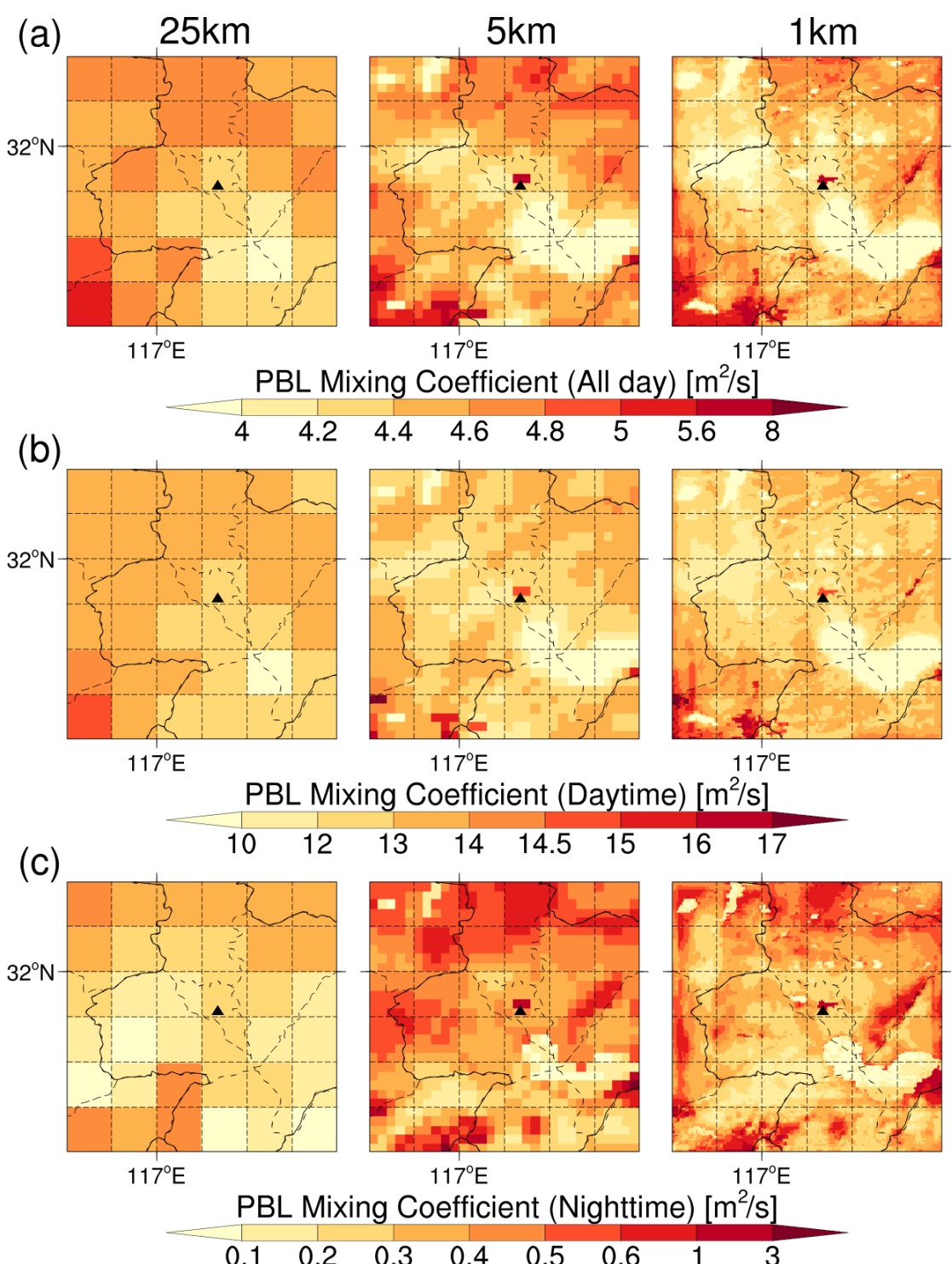

**Figure 10.** Spatial distribution of PBL mixing coefficients in the study area for 25-km
(left), 5-km (middle), and 1-km (right) resolution simulations of the whole day (top),
the daytime (middle), and the nighttime (bottom), respectively. The solid black
triangle indicates the location of the USTC site. The simulation results are from the
three sensitivity experiments.

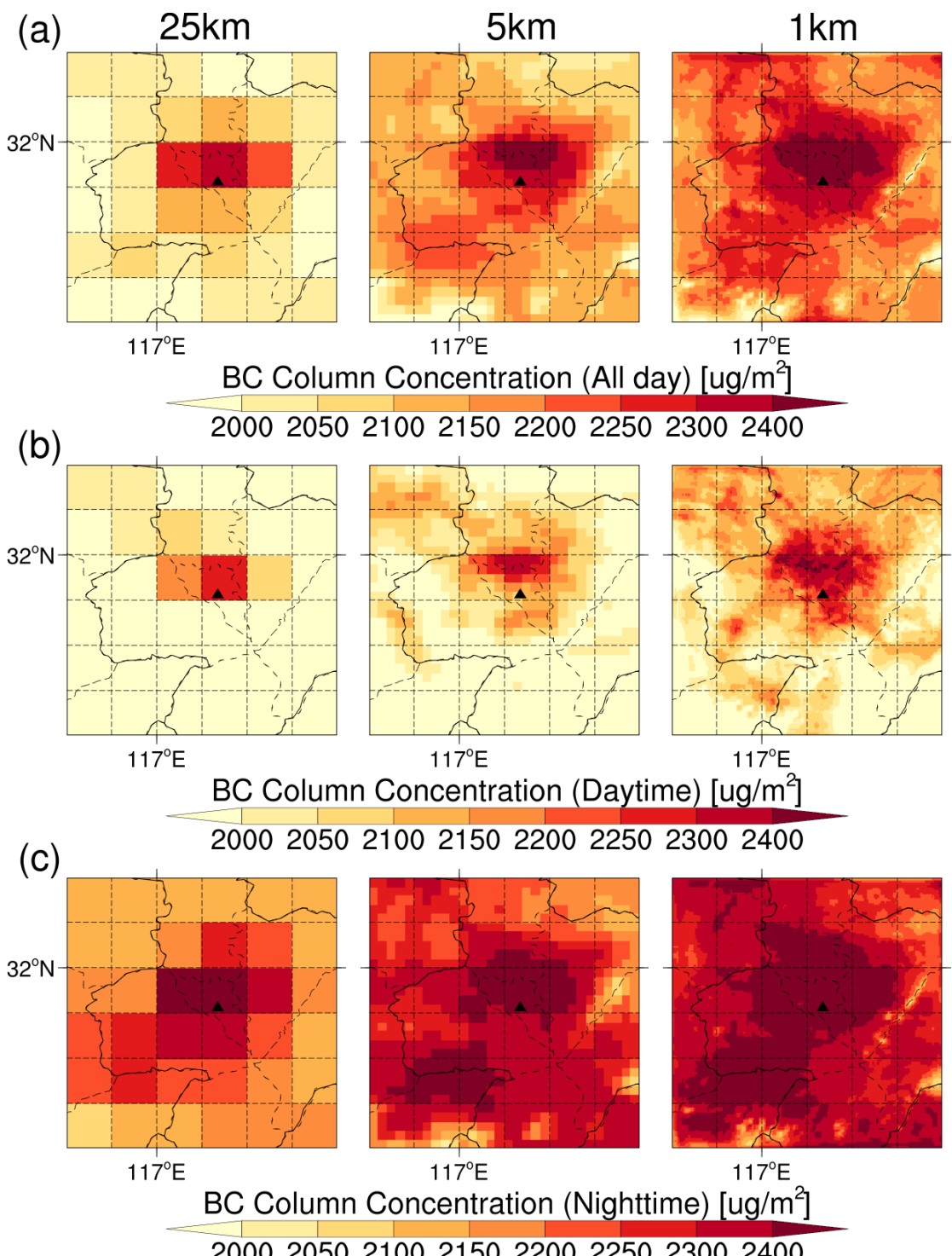

**Figure 11.** Spatial distribution of the BC column concentration in the study area for 25-km (left), 5-km (middle), and 1-km (right) resolution simulations of the whole day (top), the daytime (middle), and the nighttime (bottom), respectively. The solid black triangle indicates the location of the USTC site.

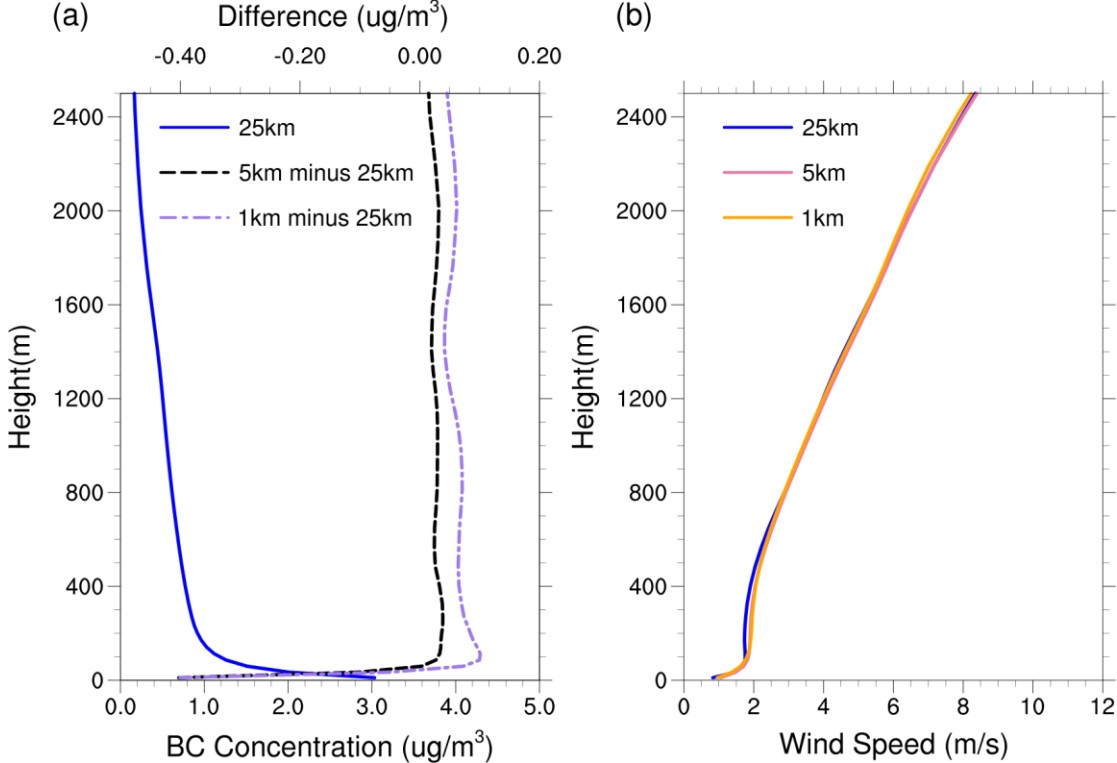


**Figure 12.** (a) Vertical profiles of BC concentrations simulated at 25-km resolution


(solid blue line), the difference between 5-km and 25-km resolutions (dashed black


line), and the difference between 1-km and 25-km resolutions (dashed purple line)


averaged over the study area for the whole day, respectively. (b) Vertical profiles of


wind speed simulated at 25-km resolution (solid blue line), 5-km resolution (solid


pink line), and 1-km resolution (solid orange line) averaged over the study area for the


whole day, respectively.












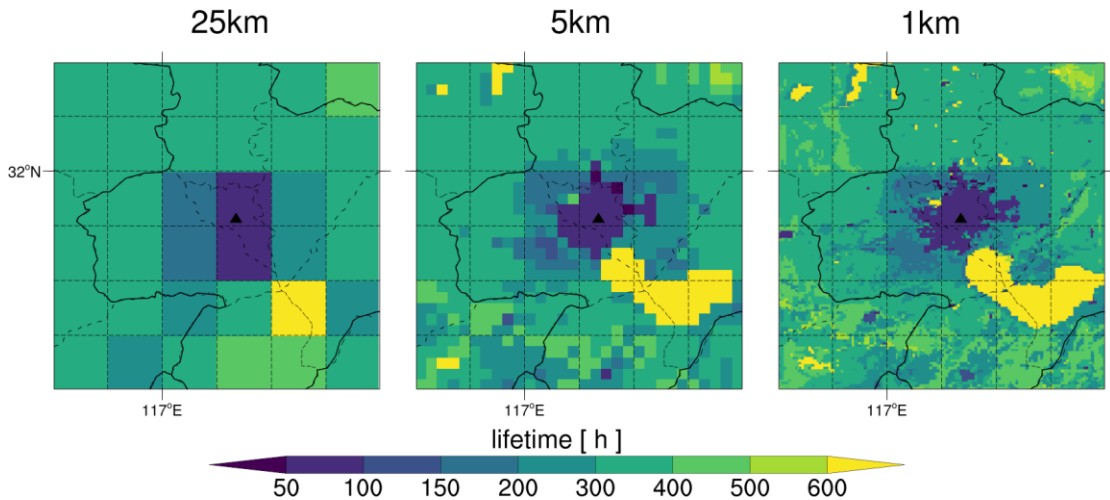


**Figure 13.** Spatial distribution of the lifetime in the study area for 25-km (left), 5-km

(middle), and 1-km (right) resolution simulations of the whole day, respectively. The

solid black triangle indicates the location of the USTC site.








