# Peer review of "Modeling urban pollutant transport at multi-resolutions: Impacts of"

_EGUsphere, 2024_

## Referee Comment (RC1)

In this paper Yang et al. use the WRF-Chem model to address the connection between horizontal model resolution and mixing strength within the atmospheric boundary layer. Previous literature on this topic had found that the diurnal cycles of modelled trace gases and aerosols show vast discrepancies to measurements, and that this may be caused by faulty parametrizations of mixing. The findings of Yang et al. align well with these previous results.

Because mixing strength appears to be a fundamental issue of the WRF-Chem model, the paper by Yang et al. is a valuable contribution to the field, and suits the scope of ACP. Moreover, the paper is well-written and delivers analyses of the model results. I wish to emphasize that I have truly appreciated the level of depth in these analyses, which reflect the authors' efforts to explain all details of their results based on physical arguments.

That being said, I would generally recommend this paper for publication. However, I do have one central concern of technical nature, which I believe must be addressed before the scientific content of the paper can be evaluated.

Unfortunately, the WRF-Chem source code that handles vertical mixing is far from transparent, and this produces some issues which also affect this article. One of the authors has previously written about the parametrization of vertical mixing (Du et al., 2020), to which the authors refer in line 69. What this paper showed was that PBL schemes, such as the YSU scheme used here, tend to produce unrealistically low mixing coefficients. This aligns well with the overestimation of surface-layer black carbon during nighttime. The problem is, that the associated mixing coefficients are not only obtained via a parametrized PBL scheme, but further manipulated in the mixing routine of the file `chem/dry_dep_driver.F`. Here, the mixing coefficients are clipped to empirically chosen thresholds of 1 $m^2$ $s^{-1}$ (over rural regions) and 2 $m^2$ $s^{-1}$ (over urban regions). The distinction between rural and urban regions is made based on the local CO emission strength. The corresponding source code comments indicate, that this clipping is indeed intended to prevent unrealistically low mixing coefficients. I have outlined a sketch of the internal procedure below:

[Figure]

In summary, if the YSU PBL scheme computes a mixing coefficient of e.g. $D = 0.5$ m$^2$ s$^{-1}$, it is set to either 1 m$^2$ s$^{-1}$ if the corresponding model pixel is considered „rural" or 2 m$^2$ s$^{-1}$, if the pixel is considered „urban", while mixing coefficients above 2 m$^2$ s$^{-1}$ remain generally unchanged.

This procedure obviously affects the paper by Yang et al. in two regards:

1) Because the distinction between rural and urban regions is made based on CO emission strength (not land-use data), the parametrization of mixing implicitly depends e.g. on the resolution of the *emission* data, not the actual model resolution.

2) The mixing coefficients that WRF-Chem uses to compute the dispersion of tracers (named $D^*$ above) are not the same as the coefficients written to the output stream (named $D$ above). Figure 7 of the paper shows $D$, which is why the values are far below the lowest clipping threshold of 1 m$^2$ s$^{-1}$. This has severe implications, because $D$ goes as low as 0.1 m$^2$ s$^{-1}$, which is an order of magnitude below $D^*$.

The authors cannot be blamed for this oversight, because the relevant source code is hard to find. Ironically, I myself have stumbled upon it after reading the paper by Du et al. (2020) and attempting to retrace the full mixing procedure of the WRF-Chem model. Our article on this issue can be found under

Kuhn, L., Beirle, S., Kumar, V., Osipov, S., Pozzer, A., Bösch, T., Kumar, R., and Wagner, T.: On the influence of vertical mixing, boundary layer schemes, and temporal emission profiles on tropospheric NO2 in WRF-Chem – comparisons to in situ, satellite, and MAX-DOAS observations, Atmos. Chem. Phys., 24, 185–217, https://doi.org/10.5194/acp-24-185-2024, 2024.

Please note, that the exact routine described above is subject to further uncertainties, including:

1) Slight changes in `chem/dry_dep_driver.F` depending on the version of the WRF-Chem model (these can be traced on the WRF github page). Particularly in commit bda151f, made on 10 January 2022, the routine was changed to exclude the lowest layer and only be effective in the layers above.

2) Possible differences to the USTC version of WRF-Chem used in the paper

In order to proceed, I would recommend the following to the authors:

1) Add a discussion on the mixing routine, perhaps in sect. 3.2.2 of the paper. I believe a plain-text explanation is sufficient. This should contain a clear outline of the exact model setup the authors have used (e.g. whether they have adapted the adjustments mentioned by Du et al., 2020, in which case the specific numerical values of the adaption should be given).

2) Modify Figure 7 to show the values of $D^*$ (see above) in addition to the existing panels. This can be achieved with a simple thresholding applied to the data already shown, but it requires that the authors know exactly which clipping thresholds were in effect during their simulation.

3) Modify all corresponding discussions referring to the low values of $D$ (particularly below 1 $m^2$ $s^{-1}$) in sect. 3.2.2 as to reflect the difference between $D$ and $D^*$. Please ensure that the line of physical reasoning (which I find to be a very strong point of the paper) remains consistent upon this modification.

---

## Referee Comment (RC2)

The research conducted by Yang et al. investigated the impact of different grid resolutions on meteorology, especially turbulent mixing, as well as its subsequent influence on air pollution transport and dispersion. Although the topic is of interest, some flaws in model validation and interpretation would undermine the scientific validity and clarity of this work and should be properly addressed before its consideration in ACP.

My major concern about this work is the validation for the model. The current validation is limited to comparisons between simulations and meteorology/BC concentration data from a single surface station. To enhance the robustness of the validation, additional stations with meteorological or air quality data should be incorporated. Furthermore, it would be beneficial to include vertical observations such as PBL height or turbulent fluxes. Comparing these with the simulations could better illustrate the mixing processes in the boundary layer. Without such data, it remains unclear whether the mixing is more accurately captured at finer resolutions.

The model settings and the interpretation of model results should be further elaborated. The vertical level settings in all three domains should be further elaborated. Are these settings sufficient to capture boundary layer turbulence accurately? Given that hourly observations for meteorology and BC are available, why is the model output provided at 3-hour intervals? This discrepancy should be addressed. The authors claim that the PBL mixing coefficient is crucial for BC surface concentration, which is closely related to land-use type and terrain. Does this imply that high-resolution surface information is more important than high-resolution atmospheric grids? This point needs further clarification. Lastly, vertical grid resolution may play a more significant role in resolving eddies of different scales. The authors should discuss this aspect in more detail.

Minor:
Line 23-28: 'higher resolution' in key points, what resolution? Vertical or horizontal? Please be specific.
Line 418: what is the diurnal variation of BC emissions?
I noticed a few spelling errors, such as 'parametrized' (should be 'parameterized') on page 17, line 474. Please review the manuscript for similar issues.

---

## Author Comment (AC1)

**Response Letter**

**Dear Editors and Reviewers:**

We sincerely thank you for facilitating the review process and providing valuable feedback. The comments received have been instrumental in enhancing our manuscript's quality and clarity.

Additionally, in response to the editorial support team's feedback, we have revised the color schemes in our maps and charts to ensure accessibility for readers with color vision deficiencies. We utilized colorblind-safe palettes from Color Brewer and implemented color schemes recommended by Crameri et al. (2020), which are specifically designed for individuals with color vision deficiencies. We also verified our figures using the Color Blindness Simulator to ensure the updated color schemes remain interpretable across various types of color blindness. We appreciate your attention to this important accessibility concern and have made these adjustments to enhance the inclusivity of our visualizations.

We have carefully considered all the comments and have made comprehensive revisions to address each point raised. Our detailed responses to the reviewers' comments are provided below.

Yours sincerely,
Zining Yang and co-authors

Reference:
Crameri, F., Shephard, G. E., and Heron, P. J.: The misuse of colour in science communication, Nature Communications, 11, https://doi.org/10.1038/s41467-020-19160-7, 2020.

**Reviewer #1**

*General comments:*

- *1. In this paper Yang et al. use the WRF-Chem model to address the connection between horizontal model resolution and mixing strength within the atmospheric boundary layer. Previous literature on this topic had found that the diurnal cycles of modelled trace gases and aerosols show vast discrepancies to measurements, and that this may be caused by faulty parametrizations of mixing. The findings of Yang et al. align well with these previous results.*

**Response:** Thank you for your thoughtful comment. We appreciate your acknowledgment of how our findings align with previous literature. As you mentioned, discrepancies in the diurnal cycles of modeled trace gases and aerosols are indeed a significant concern in atmospheric modeling, and we agree that these discrepancies may often stem from issues in turbulent mixing processes. Our work aims to further explore and refine these relationships, particularly focusing on the influence of horizontal resolution on mixing strength within the boundary layer. We hope that our findings can contribute to improving the accuracy of atmospheric simulations and better understanding the mechanisms underlying these discrepancies.

- *2. Because mixing strength appears to be a fundamental issue of the WRF-Chem model, the paper by Yang et al. is a valuable contribution to the field, and suits the scope of ACP. Moreover, the paper is well-written and delivers analyses of the model results. I wish to emphasize that I have truly appreciated the level of depth in these analyses, which reflect the authors' efforts to explain all details of their results based on physical arguments.*
  *That being said, I would generally recommend this paper for publication. However, I do have one central concern of technical nature, which I believe must be addressed before the scientific content of the paper can be evaluated.*

**Response:** Thank you very much for your kind and thoughtful comments. We are glad to hear that you consider our paper a valuable contribution to the field and that you appreciate the depth of our analysis. We have worked diligently to ensure that the model results are explained in detail based on sound physical arguments, and your recognition of this effort is greatly appreciated. We also greatly appreciate your recommendation for publication, pending the resolution of the technical concern you mentioned. We fully understand the importance of addressing this issue, and we are committed to carefully addressing your concern before proceeding further with the manuscript. We will ensure that the necessary revisions are made and that the scientific content is fully strengthened in line with your suggestions.

*Specific comments:*

- *1. Unfortunately, the WRF-Chem source code that handles vertical mixing is far from transparent, and this produces some issues which also affect this article. One of the authors has previously written about the parametrization of*

*vertical mixing (Du et al., 2020), to which the authors refer in line 69. What this paper showed was that PBL schemes, such as the YSU scheme used here, tend to produce unrealistically low mixing coefficients. This aligns well with the overestimation of surface-layer black carbon during nighttime. The problem is, that the associated mixing coefficients are not only obtained via a parametrized PBL scheme, but further manipulated in the mixing routine of the file chem/dry_dep_driver.F. Here, the mixing coefficients are clipped to empirically chosen thresholds of 1 $m^2$ $s^{-1}$ (over rural regions) and 2 $m^2$ $s^{-1}$ (over urban regions). The distinction between rural and urban regions is made based on the local CO emission strength. The corresponding source code comments indicate, that this clipping is indeed intended to prevent unrealistically low mixing coefficients. I have outlined a sketch of the internal procedure below:*

[Figure]

**Response:** Thank you for your insightful comment. We indeed noticed that this parameterization exists both in the community version of WRF-Chem and in our USTC version of WRF-Chem. While this simple parameterization is not perfect, it was designed to help mitigate the strong accumulation of air pollutants near the surface. As you pointed out, the mixing coefficients (D) used for the mixing of chemical species are clipped to empirically chosen thresholds (D*) of 1 m²/s over rural regions and 2 m²/s over urban regions. These thresholds are modified based on the differences in anthropogenic CO and primary $PM_{2.5}$ emissions between rural and urban areas.

However, it is important to clarify that this adjustment applies only to gases, not to aerosols, when using the MOSAIC or MADE/SORGAM aerosol schemes. This is because the adjustment does not couple with the aerosol PBL mixing scheme in WRF-Chem, although it could potentially be modified to be compatible with the MOSAIC scheme. In the WRF-Chem aerosol PBL mixing scheme, the original exchange coefficients (D) are still used. Therefore, for the parameterization of vertical mixing processes for atmospheric particulate matter, the WRF-Chem model continues to employ the conventional approach of directly deriving the eddy diffusion coefficient (D) from boundary layer parameterization schemes, rather than using the modified coefficients (D*). Therefore, our focus is mainly on particulate matter in this analysis and we omitted this modification for gases. The more detailed internal procedure is shown in Figure 1.

[Figure]

Figure 1. The internal procedure for mixing coefficients in the WRF-Chem model.

● *2. In summary, if the YSU PBL scheme computes a mixing coefficient of e.g. D = 0.5 m² s⁻¹, it is set to either 1 m² s⁻¹ if the corresponding model pixel is considered „rural" or 2 m² s⁻¹, if the pixel is considered „urban", while mixing coefficients above 2 m² s⁻¹ remain generally unchanged.*

**Response:** Thank you for your insightful comment. As previously discussed, for gases, if the computed mixing coefficient (D) falls below a certain threshold, it is adjusted to 1 m²/s for rural regions or 2 m²/s for urban regions. This modification prevents unrealistically low mixing coefficients. As you correctly noted, mixing coefficients exceeding 2 m²/s remain unchanged. We want to emphasize that this adjustment applies exclusively to gases, not to aerosols when the MOSAIC or MADE/SORGAM aerosol schemes are implemented. For aerosol mixing, the original computed exchange coefficients (D) are preserved since the adjustment does not integrate with the aerosol PBL mixing scheme in WRF-Chem, although it could potentially be modified to be compatible with the MOSAIC scheme. Therefore, the mixing coefficient is not modified based on the differences in anthropogenic primary $PM_{2.5}$ emissions between rural and urban areas. As discussed earlier, we have included a more comprehensive explanation of this process. In the revised manuscript (Section 4), we have added this detailed explanation: "In WRF-Chem, the mixing coefficients of chemical species are clipped to empirically chosen thresholds of 1 m²/s over rural areas and 2 m²/s over urban areas to prevent unrealistically low values. These thresholds are modified based on differences in anthropogenic CO and primary $PM_{2.5}$ emissions between rural and urban regions. Importantly, this adjustment applies exclusively to gases and not to aerosols when the MOSAIC or MADE/SORGAM aerosol schemes are used. This is because the adjustment does not couple with the aerosol PBL mixing scheme in WRF-Chem, although potential modifications could be made for compatibility with the MOSAIC or MADE/SORGAM scheme. Thus, the boundary layer mixing coefficient

for gases is implicitly influenced by emission resolution rather than directly controlled by model resolution. In this study, this treatment caused gas mixing coefficients to converge across different horizontal resolutions, preventing us from accurately assessing the impact of horizontal resolution on gas turbulent mixing. For aerosols, however, the original PBL mixing coefficients are retained, which are directly parametrized from boundary layer parameterization schemes. Therefore, our focus is mainly on particulate matter in this analysis and we omitted this modification for gases."

Additionally, we have also added relevant explanations in Section 2.2.2 and Sections 3.2.2 of the main text in the revised manuscript, as detailed in the comments below.

- *3. This procedure obviously affects the paper by Yang et al. in two regards:*

  *1) Because the distinction between rural and urban regions is made based on CO emission strength (not land-use data), the parametrization of mixing implicitly depends e.g. on the resolution of the emission data, not the actual model resolution.*

**Response:** Thank you for your insightful comment. As you correctly noted, the boundary layer mixing coefficient for gases in the WRF-Chem model is implicitly influenced by the resolution of emission data, rather than being directly controlled by the model resolution. This means that the existing adjustment process for gas mixing coefficients, which is based on CO emission strength, is not suitable for studying the impact of model resolution on the turbulent mixing of gaseous pollutants. Since the gas mixing coefficients have already been adjusted based on CO emission thresholds, they do not fully reflect the actual impact of different grid resolutions on the turbulent mixing of gases.

The main objective of our study is to investigate differences in pollutant concentrations across multi-resolutions and explore how the turbulent mixing plays as a crucial role in affecting pollutant concentrations at various resolutions. To avoid the influence of inconsistent emission data on multi-resolution simulation results in this study, we ensured that emissions at different horizontal resolutions had consistent accuracy. This allowed us to independently analyse the mechanism by which turbulent mixing influences pollutant concentration differences across various resolutions. However, after this precise treatment, due to the dependence of the gas boundary layer mixing coefficient on emission data precision, the gas mixing coefficients across different horizontal resolutions tended to converge. This does not align with our research objectives, as we were unable to fully reveal the impact of different horizontal resolutions on the turbulent mixing of gases.

In contrast, our study focuses on the turbulent mixing process of atmospheric particulate matter, where the mixing coefficient is directly calculated through boundary layer parameterization and is not adjusted based on pollutant emission data, thus avoiding limitations imposed by emission resolution. Therefore, the natural mixing of particulate matter in the model more accurately reflects the turbulent mixing process of pollutants. This approach allows us to investigate the turbulent mixing intensity of particulate matter at different resolutions and further investigate the true impact of grid resolution on pollutant mixing.

We hope this clarification better explains the challenges posed by the emission data's influence on gas mixing coefficients and how our study addresses these issues by focusing on particulate matter. In the revised manuscript (Section 2.2.2), we have added a detailed explanation:

"Although this study primarily focuses on the simulation of BC, we conducted a comprehensive validation of other air pollutants to ensure the reliability of the simulation results. However, after being initially obtained via a parameterized PBL scheme, the mixing coefficients for gases are clipped to empirically chosen thresholds of 1 $m^2$/s over rural regions and 2 $m^2$/s over urban regions, with the distinction between rural and urban regions made based on the local CO emission strength. Thus, the boundary layer mixing coefficient for gases in the WRF-Chem model is implicitly influenced by emission resolution rather than directly controlled by model resolution. Consequently, the existing adjustment process for gas mixing coefficients, which relies on CO emission strength, is unsuitable for studying the impact of model resolution on the turbulent mixing of gaseous pollutants. In contrast, the mixing coefficient of particulate matter is directly calculated through boundary layer parameterization without subsequent modifications. Thus, in this study, we limited our additional validation to $PM_{2.5}$ (fine particulate matter with aerodynamic diameters less than 2.5 µm), whose mixing processes are governed by the same resolution-dependent mechanisms as BC. Ground observations of hourly $PM_{2.5}$ surface concentrations during March 2019 were obtained from the website of the Ministry of Environmental Protection of China (MEP of China). As our study concentrates on the Hefei region, we selected 10 monitoring stations within this area for detailed analysis. These stations are marked as black triangles in Figure S5."

In the revised manuscript (Section 3.2.2), we have also added another detailed explanation:

"Previous studies have established that PBL mixing coefficients are critical determinants in air quality modeling (Du et al., 2020). In WRF-Chem, turbulent mixing within the boundary layer is partially governed by PBL mixing coefficients generated by the PBL parameterization scheme. It is worth noting that the mixing coefficients for atmospheric particulate matter and gases are two distinct variables in the current version of WRF-Chem. The boundary layer mixing coefficient for gases is initially obtained via a parameterized PBL scheme but undergoes additional modification through an empirical parameterization that enhances gas mixing based on CO emission strength (Kuhn et al., 2024). This enhancement applies to gas pollutants when using the MOSAIC aerosol scheme, as implemented in this study. Specifically, gas mixing coefficients are clipped to empirically chosen thresholds of 1 $m^2$/s over rural regions and 2 $m^2$/s over urban regions, with the distinction between rural and urban regions made based on the local CO emission strength. In contrast, the mixing coefficient of particulate matter is directly calculated through boundary layer parameterization without subsequent modifications. Our study focuses exclusively on the turbulent mixing of atmospheric particulate matter, analyzing the aerosol mixing coefficient with the default lower limit of 0.1 $m^2$/s as specified in the publicly released version of WRF-Chem. Additionally, we have not implemented the mixing coefficient adjustments proposed by Du et al. (2020), which suggest raising the lower limit of PBL mixing coefficient from 0.1 $m^2$/s to 5 $m^2$/s within the PBL. Unlike gas mixing coefficients, the particulate matter mixing coefficient is

directly calculated through boundary layer parameterization without adjustments based on pollutant emission data, thus not being limited by emission resolution. This approach allows the model to more accurately represent the natural turbulent mixing processes. Consequently, we can investigate the turbulent mixing intensity of particulate matter across different horizontal resolutions and examine the true impact of grid resolution on pollutant mixing."

*2) The mixing coefficients that WRF-Chem uses to compute the dispersion of tracers (named D\* above) are not the same as the coefficients written to the output stream (named D above). Figure 7 of the paper shows D, which is why the values are far below the lowest clipping threshold of 1 m²/s. This has severe implications, because D goes as low as 0.1 m²/s, which is an order of magnitude below D\*.*

**Response:** Thank you for your valuable comment. As discussed above, in the current version of WRF-Chem, the mixing coefficients for atmospheric particulate matter (D) and gases (D\*) are two distinct variables. The mixing coefficient for gases is adjusted based on CO emissions, with its minimum value clipped to 1 $m^2$/s. However, Figure 7 presents the mixing coefficient for particulate matter (D), which is the focus of this study, rather than the gas mixing coefficient (D\*). Therefore, the values shown in the figure are reasonable, and the observed mixing coefficients for particulate matter can indeed go as low as 0.1 m²/s, as this is the default minimum mixing coefficient in WRF-Chem. Since our study primarily addresses the mixing process of particulate matter and does not involve the gas mixing coefficient (D\*), we have not included this in the figure. We hope this clarification resolves the issue and provides a better understanding of the mixing coefficients used in our study. We also add more discussion and clarification in the revised manuscript as the response above.

● *4. The authors cannot be blamed for this oversight, because the relevant source code is hard to find. Ironically, I myself have stumbled upon it after reading the paper by Du et al. (2020) and attempting to retrace the full mixing procedure of the WRF-Chem model. Our article on this issue can be found under*
*Kuhn, L., Beirle, S., Kumar, V., Osipov, S., Pozzer, A., Bösch, T., Kumar, R., and Wagner, T.: On the influence of vertical mixing, boundary layer schemes, and temporal emission profiles on tropospheric NO2 in WRF-Chem – comparisons to in situ, satellite, and MAXDOAS observations, Atmos. Chem. Phys., 24, 185–217, https://doi.org/10.5194/acp-24-185-2024, 2024.*

**Response:** Thank you very much for your valuable comment and for pointing out the relevant source code and the article by Kuhn et al. (2024). We appreciate your understanding regarding the difficulty of locating the necessary information, and we are grateful for your effort in retracing the full mixing procedure in WRF-Chem. We have reviewed the article you referenced and will incorporate its additional insights to further improve our understanding of the vertical mixing process in the model. We recognize the importance of this information and will ensure that our manuscript is updated to reflect a more thorough consideration of the vertical mixing procedure in future versions of the model. We have now cited the Kuhn et al. (2024) article

in Section 3.2.2 of the revised manuscript, as it provides valuable insights into the vertical mixing process and its influence on model result: "The boundary layer mixing coefficient for gases is initially obtained via a parameterized PBL scheme but undergoes additional modification through an empirical parameterization that enhances gas mixing based on CO emission strength (Kuhn et al., 2024)."

- *5. Please note, that the exact routine described above is subject to further uncertainties, including:*

  *1) Slight changes in chem/dry_dep_driver.F depending on the version of the WRF-Chem model (these can be traced on the WRF github page). Particularly in commit bda151f, made on 10 January 2022, the routine was changed to exclude the lowest layer and only be effective in the layers above.*

  *2) Possible differences to the USTC version of WRF-Chem used in the paper*

**Response:** Thank you for your insightful comment. While there are several modifications between the publicly released WRF-Chem version and the USTC version, these changes are not directly related to the focus of our study and thus do not affect our results. Therefore, we have not incorporated these modifications into our analysis. Specifically, the computation of mixing coefficients for both atmospheric particulate matter and gases remains similar in both versions. The gas mixing coefficient, originally derived from the parameterized PBL scheme, undergoes further modification through an empirical parameterization based on CO emission strength, enhancing gas mixing. This adjustment is applied to gas pollutants when the MOSAIC aerosol scheme is used, with the mixing coefficients clipped to thresholds of 1 $m^2$/s for rural regions and 2 $m^2$/s for urban regions. In contrast, the mixing coefficient for particulate matter is directly calculated through boundary layer parameterization without additional modifications. We acknowledge that the USTC version does not include the changes made in commit bda151f (January 10, 2022), which modified the routine to exclude the lowest layer and apply only to layers above. However, since our study primarily focuses on the turbulent mixing of atmospheric particulate matter rather than gas PBL mixing coefficients specifically, this difference does not impact our findings. The computations and processes directly relevant to our research remain highly consistent between versions, ensuring the validity of our results.

- *6. In order to proceed, I would recommend the following to the authors:*

  *1) Add a discussion on the mixing routine, perhaps in sect. 3.2.2 of the paper. I believe a plain-text explanation is sufficient. This should contain a clear outline of the exact model setup the authors have used (e.g. whether they have adapted the adjustments mentioned by Du et al., 2020, in which case the specific numerical values of the adaption should be given).*

**Response:** Thank you for your valuable suggestion for raising this critical point regarding the clarity of the mixing routine in our model setup. In response to your recommendation, we have added a detailed discussion on the mixing routine in Section 3.2.2 of the revised manuscript to explicitly outline the mixing routine configuration. These additions ensure full transparency

in our methodology and directly address the reviewer's request for a "plain-text explanation" of the mixing routine. The updated section also clarifies why our approach differs from Du et al.'s (2020) recommendations, thereby strengthening the interpretability of our results. This section includes a clear outline of the exact model setup we used: "Previous studies have established that PBL mixing coefficients are critical determinants in air quality modeling (Du et al., 2020). In WRF-Chem, turbulent mixing within the boundary layer is partially governed by PBL mixing coefficients generated by the PBL parameterization scheme. It is worth noting that the mixing coefficients for atmospheric particulate matter and gases are two distinct variables in the current version of WRF-Chem. The boundary layer mixing coefficient for gases is initially obtained via a parameterized PBL scheme but undergoes additional modification through an empirical parameterization that enhances gas mixing based on CO emission strength (Kuhn et al., 2024). This enhancement applies to gas pollutants when using the MOSAIC aerosol scheme, as implemented in this study. Specifically, gas mixing coefficients are clipped to empirically chosen thresholds of 1 m²/s over rural regions and 2 m²/s over urban regions, with the distinction between rural and urban regions made based on the local CO emission strength. In contrast, the mixing coefficient of particulate matter is directly calculated through boundary layer parameterization without subsequent modifications. Our study focuses exclusively on the turbulent mixing of atmospheric particulate matter, analyzing the aerosol mixing coefficient with the default lower limit of 0.1 m²/s as specified in the publicly released version of WRF-Chem. Additionally, we have not implemented the mixing coefficient adjustments proposed by Du et al. (2020), which suggest raising the lower limit of PBL mixing coefficient from 0.1 m²/s to 5 m²/s within the PBL. Unlike gas mixing coefficients, the particulate matter mixing coefficient is directly calculated through boundary layer parameterization without adjustments based on pollutant emission data, thus not being limited by emission resolution. This approach allows the model to more accurately represent the natural turbulent mixing processes. Consequently, we can investigate the turbulent mixing intensity of particulate matter across different horizontal resolutions and examine the true impact of grid resolution on pollutant mixing."

Reference:

Du, Q., Zhao, C., Zhang, M., Dong, X., Chen, Y., Liu, Z., Hu, Z., Zhang, Q., Li, Y., Yuan, R., and Miao, S.: Modeling diurnal variation of surface PM2.5 concentrations over East China with WRF-Chem: impacts from boundary-layer mixing and anthropogenic emission, Atmospheric Chemistry and Physics, 20, 2839-2863, https://doi.org/10.5194/acp-20-2839-2020, 2020.

Kuhn, L., Beirle, S., Kumar, V., Osipov, S., Pozzer, A., Boesch, T., Kumar, R., and Wagner, T.: On the influence of vertical mixing, boundary layer schemes, and temporal emission profiles on tropospheric NO2 in WRF-Chem - comparisons to in situ, satellite, and MAX-DOAS observations, Atmospheric Chemistry and Physics, 24, 185-217, https://doi.org/10.5194/acp-24-185-2024, 2024.

*2) Modify Figure 7 to show the values of D\* (see above) in addition to the existing panels. This can be achieved with a simple thresholding applied to the data already shown, but it requires that the authors know exactly which clipping thresholds were in effect during their simulation.*

**Response:** Thank you for your helpful suggestion. We would like to clarify that our study focuses specifically on the mixing coefficient for atmospheric particulate matter (D), which is directly calculated through boundary layer parameterization. The gas mixing coefficient, referred to as D\*, is not included in our analysis because our research addresses the turbulent mixing of particulate matter, not gases. Since D\* is not part of the analysis in this study, we do not display its distribution. The current figure already presents the relevant data for the particulate matter mixing coefficient (D), in alignment with our research scope. We hope this clarifies the reason why D\* is not included, and we appreciate your understanding. If you have any further suggestions, we would be happy to address them.

*3) Modify all corresponding discussions referring to the low values of D (particularly below 1 $m^2$ $s^{-1}$) in sect. 3.2.2 as to reflect the difference between D and D\*. Please ensure that the line of physical reasoning (which I find to be a very strong point of the paper) remains consistent upon this modification.*

Response: Thank you for your helpful suggestion. As mentioned earlier, this study focuses on the mixing coefficient for atmospheric particulate matter (D), not the gas mixing coefficient (D\*) that you referenced. The low values of D (particularly below 1 $m^2$/s) presented in Section 3.2.2 accurately reflect the mixing of particulate matter and align with the physical reasoning in the manuscript, especially regarding the boundary layer mixing process for aerosols. Since D\* is not the focus in this study, we have carefully revised the relevant discussions in Section 3.2.2 to clarify that the low values observed are associated with the particulate matter mixing coefficient (D), not D\*. This revision maintains consistency in our physical reasoning, as we continue to focus on turbulent mixing of particulate matter, which is directly influenced by boundary layer parameterization.

**Reviewer #2**

*General comments:*

● *The research conducted by Yang et al. investigated the impact of different grid resolutions on meteorology, especially turbulent mixing, as well as its subsequent influence on air pollution transport and dispersion. Although the topic is of interest, some flaws in model validation and interpretation would undermine the scientific validity and clarity of this work and should be properly addressed before its consideration in ACP.*

**Response:** Thank you for your thorough evaluation of our manuscript and for identifying areas needing improvement. We fully agree that model validation and interpretation are critical to ensuring the scientific rigor and clarity of our work. Your constructive feedback has provided valuable direction for our revisions. We are committed to addressing the clarifications you have highlighted and have made the necessary revisions to enhance both the clarity and quality of the paper. Below, we outline our responses to each of your points to ensure that the manuscript meets the standards for publication in ACP.

*Specific comments:*

● *1. My major concern about this work is the validation for the model. The current validation is limited to comparisons between simulations and meteorology/BC concentration data from a single surface station. To enhance the robustness of the validation, additional stations with meteorological or air quality data should be incorporated.*

**Response:** We sincerely appreciate your constructive feedback regarding the validation of our model. We fully agree that expanding validation beyond a single surface station is critical for ensuring the robustness of the simulation results. In response to this concern, we have significantly enhanced the validation framework by incorporating multi-site meteorological observations and multi-pollutant air quality data from additional monitoring stations. Below we provide a detailed summary of the revisions made to address this critical issue.

1. Enhanced Meteorological Data Validation

As you rightly pointed out, validation using data from a single station is insufficient to establish the model's robustness. To address this, we have integrated data from four additional Automatic Weather Stations (AWS) located in the Hefei region (marked by purple solid dots in Figure S5), thereby broadening the scope of the meteorological validation. These AWS stations provide high-quality, hourly data for several key meteorological variables, including 2 m temperature and 10 m wind speed. The data undergoes strict quality control (QC) by local meteorological bureaus, following World Meteorological Organization guidelines (Estevez et al., 2011), ensuring their reliability for comparison with model simulations. We have updated Section 2.2.1 of the manuscript to include this new data source: "Additionally, we employed meteorological data from automatic weather stations (AWSs), which were established based on the operational standards issued by the China Meteorological Administration (CMA, 2018). The hourly data underwent quality control (QC) by local meteorological bureaus of Anhui,

following World Meteorological Organization guidelines (Estevez et al., 2011). The QC included checks of consistency, such as internal, temporal-spatial, and climatic range validations. These QC data were used to determine daily mean, minimum, and maximum meteorological variables. The AWSs recorded various parameters, including air temperature (T, °C), wind speed (U, m/s), air pressure (P, Pa), and wind direction. In this study, we focus on the 3-hourly 2 m temperature and 10 m wind speed obtained from four AWS stations located in the study region. The four AWS sites are marked by purple solid dots in Figure S5."

We carefully analyzed the simulation results for these new meteorological data from AWS and found that the model performs consistently across different resolutions, as shown in Figure S6. Specifically, the comparison between simulated and observed 10-m wind speeds and 2-m temperatures from these stations generally showed good agreement, although there were occasional discrepancies in some time periods. For example, the wind speed was sometimes overestimated or underestimated, but the overall trends, including diurnal variations, were well captured by the model. This expanded validation offers a more comprehensive evaluation of the model's performance across a range of meteorological variables and geographical locations, addressing the concern about the limited validation in the initial manuscript. Section 3.1 has been updated to include a detailed comparison between the simulated and observed meteorological variables across the four AWS stations: "Additionally, Figure S6 displays the time series of observed and simulated meteorological variables averaged across four AWS stations in the study region. Figure S6a presents a comparison of 10-m wind speed simulated at three different resolutions, revealing generally consistent results with observations. The overall pattern is similar to that observed at the single USTC station, characterized by a clear diurnal variation with higher wind speeds during daytime and lower speeds at night. However, simulations at all three resolutions occasionally deviate from observations. For example, on March 11, the 5 km and 1 km resolution models overestimate wind speed at approximately 7 m/s compared to the observed 4 m/s. Conversely, on March 14 during the daytime, all three resolutions underestimate wind speed, simulating around 2 m/s against an observed value of 4 m/s. Figure S6b compares the simulated 2-m temperatures across three resolutions with observational data. The simulated temperatures are remarkably similar across all resolutions and show strong correlation with observations throughout most of the study period. Only a few outliers were noted, which minimally impact the overall pattern. For example, all resolution models overestimate temperature at noon on March 20, simulating approximately 28°C while the observed temperature is only about 20°C."

2. Expanded Air Quality Data Validation

Regarding the validation of air pollutants, we agree with your suggestion to include additional observational data. In response, we incorporated data from 10 monitoring stations within the Hefei region, obtained from the website of the Ministry of Environmental Protection of China (MEP of China). These stations, marked as black triangles in Figure S5, provide hourly surface concentration data for common pollutants. However, the boundary layer mixing coefficient for gases in the WRF-Chem model is implicitly influenced by emission resolution rather than directly controlled by model resolution. As another reviewer noted, the associated mixing coefficients are not only obtained via a parametrized PBL scheme, but further

manipulated in the mixing routine. The mixing coefficients are clipped to empirically chosen thresholds of 1 m²/s over rural regions and 2 m²/s over urban regions. The distinction between rural and urban regions is made based on the local CO emission strength. The corresponding source code comments indicate that this clipping is indeed intended to prevent unrealistically low mixing coefficients. This treatment caused gas mixing coefficients across different horizontal resolutions to converge in our study, preventing us from effectively revealing the true impact of varying horizontal resolutions on gas turbulent mixing. Therefore, the existing adjustment process for gas mixing coefficients is unsuitable for our research objectives of studying resolution impacts on turbulent mixing. Thus, in this study, we limited our additional validation to $PM_{2.5}$ (fine particulate matter with aerodynamic diameters less than 2.5 µm), whose mixing processes are governed by the same resolution-dependent mechanisms as BC.

By comparing model simulations with $PM_{2.5}$ surface concentrations, we can ensure that our model accounts for a broader range of air quality parameters and offers a more thorough evaluation of its performance, despite the primary focus on black carbon (BC) simulation. We have updated Section 2.2.2 to introduce this new observational pollutant data: "Although this study primarily focuses on the simulation of BC, we conducted a comprehensive validation of other air pollutants to ensure the reliability of the simulation results. However, after being initially obtained via a parameterized PBL scheme, the mixing coefficients for gases are then clipped to empirically chosen thresholds of 1 m²/s over rural regions and 2 m²/s over urban regions, with the distinction between rural and urban regions made based on the local CO emission strength. Thus, the boundary layer mixing coefficient for gases in the WRF-Chem model is implicitly influenced by emission resolution rather than directly controlled by model resolution. Consequently, the existing adjustment process for gas mixing coefficients, which relies on CO emission strength, is unsuitable for studying the impact of model resolution on the turbulent mixing of gaseous pollutants. In contrast, the mixing coefficient of particulate matter is directly calculated through boundary layer parameterization without subsequent modifications. Thus, in this study, we limited our additional validation to $PM_{2.5}$ (fine particulate matter with aerodynamic diameters less than 2.5 µm), whose mixing processes are governed by the same resolution-dependent mechanisms as BC. Ground observations of hourly $PM_{2.5}$ surface concentrations during March 2019 were obtained from the website of the Ministry of Environmental Protection of China (MEP of China). As our study concentrates on the Hefei region, we selected 10 monitoring stations within this area for detailed analysis. These stations are marked as black triangles in Figure S5."

We compared the simulated diurnal variation of $PM_{2.5}$ surface concentrations across three horizontal resolutions with observations from the 10 monitoring stations, as shown in Figure S10. Our results indicate that the model generally captures the diurnal variations of $PM_{2.5}$ surface concentrations, with some discrepancies. Specifically, the model tends to slightly underestimate $PM_{2.5}$ during the daytime and overestimate it during the nighttime across three resolutions. However, higher spatial resolutions (5 km and 1 km) notably improved nocturnal simulations, demonstrating the benefit of fine grid resolutions in capturing localized pollution events. We further analyzed the diurnal variation of surface concentrations of four $PM_{2.5}$ components (sulfate ($SO_4^{2-}$), nitrate ($NO_3^-$), other inorganics (OIN), and organic carbon (OC)) averaged over 10 MEP sites in

Hefei, as shown in Figure S11. In Section 3.2.1 of the revised manuscript, we have added a detailed comparison of the simulated and observed $PM_{2.5}$ surface concentrations at these 10 MEP stations:

"To verify the accuracy and comprehensiveness of the simulation results, we further analyzed the diurnal variation of $PM_{2.5}$ surface concentrations. Figure S10 illustrates the diurnal variation of simulated $PM_{2.5}$ surface concentrations across multi-resolutions compared with observations. The diurnal pattern of $PM_{2.5}$ closely resembles that of BC, characterized by higher concentrations at night and lower concentrations during daytime. Across all resolutions, the model slightly underestimates daytime $PM_{2.5}$ surface observations while overestimating nighttime values. Notably, increased horizontal resolution substantially improves nocturnal simulations. The 25 km resolution simulation generates an anomalous midnight peak (105 μg/m³), resulting in a +61% bias, whereas the 5 km and 1 km resolutions substantially mitigate these deviations to approximately 30%. To further examine the contribution of each $PM_{2.5}$ component to the diurnal variation across multi-resolutions, Figure S11 shows the diurnal variations of four $PM_{2.5}$ constituents (sulfate ($SO_4^{2-}$), nitrate ($NO_3^-$), OIN, and organic carbon (OC)) averaged over 10 MEP sites in Hefei. Significant differences emerge in the diurnal variations of these components across multi-resolution simulations. Specifically, the surface concentrations of $NO_3^-$, OIN, and OC exhibit a consistent diurnal pattern, with lower concentrations during daytime and higher concentrations at night. As resolution increases from 25 km to 5 km and 1 km, the simulated components surface concentrations decrease, aligning more closely with observations.

The total concentrations of $PM_{2.5}$ and its components demonstrate significant sensitivity to horizontal resolutions. Coarse resolution simulations underestimate turbulent mixing capacity, resulting in overestimated concentrations. Higher resolution simulations more accurately capture vertical mixing within the PBL. For secondary particles such as sulfates and nitrates, formation rates depend heavily on local precursor substance concentrations ($SO_2$, $NO_x$). Higher resolution simulations may enable more realistic representation of precursor substance diffusion, leading to reduced local concentration gradients and consequently slower secondary aerosol formation rates. Additionally, variations in $PM_{2.5}$ surface concentrations across multi-resolutions may also stem from complex secondary particle generation mechanisms. For instance, liquid-phase oxidation of sulfates in clouds is sensitive to local cloud water distribution, with higher resolutions better capturing small-scale cloud structures that potentially alter sulfate formation efficiency. The formation of ammonium nitrate ($NH_4NO_3$) is particularly sensitive to temperature and humidity variations. At higher resolutions, temperature and humidity gradients induced by urban heat island effects or topographical variations can be more realistically simulated, influencing the distribution of gaseous nitric acid ($HNO_3$) and particulate nitrate ($NO_3^-$). Dry deposition processes may also contribute to resolution-dependent variations, as local differences in surface roughness (including buildings and vegetation) become more apparent at higher resolutions, directly affecting particulate deposition velocity rates. Overall, the simulation results for major air pollutants fall within a reasonable error range compared to observational data, confirming the reliability of the model for this study."

We also added this additional validation in Section 4 of the revised manuscript: "In addition, the diurnal variation of $PM_{2.5}$

surface concentrations simulated at different resolutions follows the same trend as the observed concentrations at the national monitoring sites, with slight underestimation during daytime and overestimation at night."

We believe that the inclusion of additional meteorological and pollutant data, as well as the expanded validation analyses, significantly enhance the reliability of our model and its representation of real-world conditions. These improvements address the concerns raised regarding the limited validation in the original manuscript and offer a more comprehensive and rigorous evaluation of the model's performance.

On the other hand, while model evaluation constitutes an essential component of research, we did not allocate extensive resources to detailed model validation in this study, given our use of WRF-Chem - a well-established and thoroughly validated tool widely adopted for climate and air quality simulations. Our primary objective is to leverage this model to investigate the underlying physical and chemical mechanisms, rather than to conduct a comprehensive performance assessment.

[Figure]

Figure S5. Spatial distributions of 4 MEP and 10 AWSs observational sites used in this study. Solid black triangles indicate MEP monitoring sites, and purple solid dots indicate AWSs locations.

[Figure]

Figure S6. Time series averaged over 4 AWS sites in Hefei of observed (black dot) and simulated wind speed at 8 m (top, unit: m s-1) and temperature at 2 m (middle, unit: °C) for 25-km (solid blue line) resolution, 5-km (solid pink line) resolution, and 1-km (solid orange line), respectively.

[Figure]

Figure S10. Diurnal variation of PM$_{2.5}$ surface concentrations within 24 h averaged over 10 MEP sites in Hefei during the study period for 25-km (solid blue line), 5-km (solid pink line), and 1-km (solid orange line) resolution simulations and observations (black dot). Both the simulated results and observations are sampled at the model output frequency, i.e., 3-hourly.

[Figure]

Figure S11. Diurnal variation of (a) $SO_4^{2-}$, (b) $NO_3^-$, (c) OIN, and (d) OC surface concentrations within 24 h averaged over 10 MEP sites in Hefei during the study period for 25-km (solid blue line), 5-km (solid pink line), and 1-km (solid orange line) resolution simulations. The simulated results are sampled at the model output frequency, i.e., 3-hourly.


  - *Comparing these with the simulations could better illustrate the mixing processes in the boundary layer. Without such data, it remains unclear whether the mixing is more accurately captured at finer resolutions.*

**Response:** We sincerely appreciate your insightful suggestion regarding the inclusion of vertical observations (e.g., PBL height or turbulent fluxes) to validate boundary layer mixing processes. Below, we clarify the theoretical framework guiding our approach, address the current limitations, and outline future directions to advance this line of inquiry.

In our model, planetary boundary layer height (PBLH) does not directly represent the turbulence mixing intensity but is

a diagnostic quantity calculated through the boundary layer parameterization scheme. According to the study by Du et al. (2020), the daily variation in surface particulate matter concentration is primarily influenced by the boundary layer mixing coefficient (Kh) rather than PBLH. Previous studies often established a direct relationship between the daily variation of PBLH and pollutant concentrations. However, in reality, the daily variation of PBL mixing coefficients is much more significant than that of PBLH. During the day, when the PBL mixing coefficient is large, emitted pollutants are well mixed and distributed up to the PBL height. However, at night, due to the lower PBL mixing coefficient, atmospheric particulate matter and its precursors tend to stay near the ground and are trapped in the shallow boundary layer, causing higher surface pollutant concentrations. Thus, the PBL mixing coefficient at night becomes the primary determinant of pollutant concentration changes, while PBLH cannot fully reflect the turbulence mixing intensity in the model. Therefore, we believe that comparing the simulated PBLH with observations cannot demonstrate well whether a finer resolution can more accurately capture turbulence mixing.

Furthermore, through kilometer-scale high-resolution numerical experiments, we found that traditional parameterization schemes exhibit systematic bias in representing turbulence mixing intensity, and the boundary layer mixing coefficient does not fully reflect the turbulence mixing intensity in the model. As the resolution increases, the explicitly resolved turbulence mixing intensity in the model increases. Therefore, the total turbulence mixing intensity in the model should be the sum of the boundary layer mixing coefficient given by the parameterization scheme and the directly resolved turbulence mixing intensity.

Regarding your comment about the lack of observational data for turbulence mixing intensity, our study focuses on the Hefei city, a typical mega-city of China, which currently lacks direct observational data for overall turbulence mixing intensity. However, the analysis of turbulence mixing intensity with observations has become an important entry point in our research. While we currently lack direct observational data, we believe this analytical approach holds great potential for understanding the variations in pollutant concentrations in the Hefei area. The primary processes affecting BC concentration are emissions and turbulence mixing, with BC concentration being almost unaffected by chemical processes. In this study, we ensured that pollutant emission details were consistent across all resolutions, effectively isolating the interference of emission heterogeneity on concentration variations. Therefore, under the condition of emission homogenization, the hypothesis that vertical turbulence mixing is the core mechanism driving surface-level BC concentration fluctuations is a reasonable assumption. For future work, we plan to conduct further analytical research in urban areas with pollutant observation towers to validate the assumptions in this study and supplement more field observational data to support the results of our model.


**Response:** We sincerely appreciate your insightful feedback regarding the elaboration of model settings and vertical level configurations. The concerns about whether our vertical resolution settings adequately capture boundary layer turbulence are critically important for validating the model's reliability. We explicitly emphasize that identical vertical layer distributions are maintained across all three horizontal resolutions (25 km, 5 km, and 1 km), ensuring comparability in turbulent mixing across different horizontal resolutions. In this study, three resolution simulations employ 50 vertical layers (spanning 15 km vertically), with 30 layers allocated below 2 km to resolve the core regions of turbulent mixing. This enhanced vertical configuration ensures that we can resolve the fine details of the Planetary Boundary Layer (PBL) turbulence, which plays a crucial role in the dispersion and vertical distribution of pollutants. The vertical layers are strategically designed with varying thicknesses to provide adequate resolution across different heights. The distribution of layers designed with 7 layers below 200 meters, with each layer approximately 20 meters in height; 3 layers between 200 and 300 meters, each approximately 30 meters in height; 8 layers between 300 and 1000 meters, with each layer approximately 80 meters in height. While the near-surface vertical resolution does not adopt an ultra-fine grid (e.g., 10 m layer thickness), a stretched grid design (gradually increasing resolution near the surface) ensures reasonable representation of surface heat fluxes and turbulent kinetic energy (TKE) transport. Specifically, Jiang et al. (2024) and Jiang and Hu (2023) demonstrated that increasing vertical layers under stable boundary conditions enhances the stability of vertical structures, thereby better resolving boundary layer turbulence. Our configuration (30 layers below 2 km) comprehensively covers the mixed layer development and captures key turbulent processes (e.g., entrainment and surface flux exchange) through layer densification. The finer vertical resolution in our simulations contributes to a more stable vertical distribution of pollutants and enhances the overall performance of the model in simulating dispersion processes.

In the revised Methods section (Section 2.1.2), we have added a detailed explanation of the vertical layer configuration: "On the other hand, the vertical configuration within the PBL is also crucial for accurately modeling pollutant dispersion. To better resolve the PBL structure and mixing processes, we implemented a finer vertical resolution within the PBL. Identical vertical layer distributions are maintained across all three horizontal resolutions (25 km, 5 km, and 1 km), ensuring direct comparability of turbulent mixing across different horizontal resolutions. A total of 50 terrain-following vertical eta-layers extending from the surface to approximately 15 km were used in all three resolution simulations, with 30 layers distributed below 2 km above the ground to describe the atmospheric boundary structure in detail. The vertical layer was strategically designed with 7 layers below 200 meters (each approximately 20 meters in height), 3 layers between 200 and 300 meters (each

about 30 meters in height), and 8 layers between 300 and 1000 meters (each approximately 80 meters in height). This configuration comprehensively captures mixed layer development and key turbulent processes (e.g., entrainment and surface flux exchange) through layer densification, which is sufficient to capture PBL turbulent mixing. Jiang et al. (2024) and Jiang and Hu (2023) have demonstrated that the number of model vertical layers primarily influences vertical distribution, with more vertical grid layers producing a more stable vertical structure under stable boundary conditions that better resolves boundary layer turbulence."


- ● *4. Given that hourly observations for meteorology and BC are available, why is the model output provided at 3-hour intervals? This discrepancy should be addressed.*

**Response:** We appreciate your astute observation regarding the temporal resolution of model outputs. Now we provide a detailed justification for this design choice and confirm its validity for the study's objectives. First, the outputs of pollutant concentrations in our experiments are sampled every 3-hour to reduce disk storage requirements and increase the computational speed. Hourly output data would provide higher time resolution but significantly increase storage demands. Given that we ran simulations at multi-resolutions (25 km, 5 km, and 1 km), storing hourly data would have resulted in an enormous volume of data storage. To ensure computational and storage efficiency, we opted for a 3-hour output interval. Second, while hourly observational data are available, the primary goal of our study is to analyze the daily variations of pollutants (e.g., BC), rather than focusing on precise hourly variation comparisons. A 3-hour output interval is sufficient to capture the daily trends, and we believe it provides an effective representation of the daily variability without losing significant detail. Therefore, when comparing modeling results and observations, hourly observations are sampled at the model output frequency, i.e., 3-hourly.

In the revised manuscript (Section 2.2.2), we have also added another detailed explanation for this design choice:

"While hourly observations for both meteorology and pollutants are available, model outputs are provided at 3-hour intervals to balance computational efficiency and storage requirements. Hourly output data would provide higher time resolution but significantly increase storage demands. Given that we ran simulations at multi-resolutions (25 km, 5 km, and 1 km), hourly outputs would have generated prohibitively large data volumes. On the other hand, this 3-hour output interval remains sufficient for our primary research objective of analyzing daily pollutant variations (particularly BC) rather than precise hourly comparisons. We believe this approach effectively captures daily variability patterns without losing essential

detail. For direct comparisons, hourly observations were sampled to match our 3-hour model output intervals."

● *5. The authors claim that the PBL mixing coefficient is crucial for BC surface concentration, which is closely related to land-use type and terrain. Does this imply that high-resolution surface information is more important than high-resolution atmospheric grids? This point needs further clarification.*

**Response:** We appreciate your insightful comment. We agree that the role of high-resolution surface information and high-resolution atmospheric grids in parameterizing the boundary layer (PBL) mixing coefficient requires further clarification.

In our study, both high-resolution surface information and atmospheric grid resolution contribute to the parameterization of the boundary layer mixing coefficient. In regions with substantial variations in land use and terrain height, high-resolution surface information plays a more dominant role in parameterizing the PBL mixing coefficient. For example, in areas with complex terrain or varying land-use types, more detailed surface information (such as more accurate land use classification at higher resolutions) significantly enhances the calculation of the PBL mixing coefficient. This is because the parameterization of the mixing coefficient is directly influenced by the surface roughness, which is highly sensitive to land-use and terrain characteristics. On the other hand, if high-resolution simulations do not use refined high-resolution surface information (for example, the same low-resolution surface data is used across all three resolutions), the increase in the boundary layer mixing coefficient is mainly due to the higher atmospheric grid resolution. In areas with relatively flat terrain or homogeneous land-use types, the increase in the boundary layer mixing coefficient primarily stems from the higher atmospheric grid resolution. Additionally, high-resolution atmospheric grids help resolve finer atmospheric processes, such as small-scale turbulent eddies. In conclusion, land use and terrain accuracy dominate the simulation of mixing coefficients in regions with significant surface heterogeneity, while high-resolution grids enhance the mixing effect in homogeneous surface regions through numerical and physical mechanisms.

To address this issue more comprehensively, we have conducted an additional set of sensitivity experiments. These experiments utilized the default United States Geological Survey (USGS) land use data in the Weather Research and Forecasting (WRF) model across various resolutions to isolate the effects of grid resolution under uniform land-use conditions, as shown in Figure S2. The default USGS data in WRF's geographical static database represents Chinese land use patterns before the 2000s and reflects land use distribution prior to China's significant urbanization. Consequently, the land use data types have minor variations and remained generally consistent across all three resolutions in the sensitivity experiment. The results clearly demonstrate that in flat urban areas, where land-use types and terrain are nearly uniform across different resolutions, the increase in PBL mixing coefficient is primarily driven by the finer grid resolution. In contrast, in areas with significant terrain height variations, such as rural or hilly regions, the boundary layer mixing coefficient is more sensitive to surface roughness and topographical details, which are better captured at higher resolutions. We have added the specific setup of this additional set of sensitivity experiments with different resolutions in section 2.1.2 in the revised manuscript: "The land

cover dataset is derived from a 1 km horizontal resolution dataset for China (Zhang et al., 2021). The land use categories follow the United States Geological Survey's (USGS) 24-category classification, and the dataset is based on China's land cover conditions as of 2015. This provides a more accurate representation of current land cover, particularly for eastern China, which has experienced intensive urban expansion since the 2000s. Figure 1b shows the land cover data at different resolutions, with detailed descriptions of the legend and land cover classes provided in Table S1. This set of simulations is referred to as the baseline experiment. With the exception of part of Section 3.2.3, all other analyses in this study are based on the results of these baseline experiments. Moreover, to explore the differences in turbulent mixing simulated at multi-resolutions under consistent land use conditions across various resolutions, we conducted an additional set of sensitivity experiments referred to as the sensitivity experiment. The sensitivity experiment was identical to the baseline experiment, except it used the default USGS land use category data in WRF. Notably, this default USGS data in WRF's geographical static database represents Chinese land use patterns before the 2000s, as shown in Figure S2. This default dataset reflects land use distribution prior to China's significant urbanization. Consequently, the land use data types have minor variations and remained generally consistent across all three resolutions in the sensitivity experiment."

Our results show that in regions with heterogeneous surface types (such as urban areas), high-resolution surface data is critical for accurately parameterizing the PBL mixing coefficient. In contrast, in homogeneous regions or regions with relatively flat terrain, the high-resolution atmospheric grid helps resolve finer-scale turbulent features, further enhancing the mixing effect. In conclusion, while high-resolution surface information is paramount in regions with significant land-use and terrain variations, high-resolution atmospheric grids also play a crucial role in enhancing the mixing effect, especially in areas with more gradual surface changes. Therefore, both components are important, but the relative importance depends on the surface heterogeneity and the complexity of the terrain. We have elaborated on these findings in the newly added Section 3.2.3, which discusses the impact of land use 
[revised manuscript text omitted]

- *6. Lastly, vertical grid resolution may play a more significant role in resolving eddies of different scales. The authors should discuss this aspect in more detail.*

**Response:** We appreciate the reviewer's insightful comment regarding the potential role of vertical grid resolution in resolving eddies of different scales. We agree that vertical resolution can indeed influence the representation of turbulent processes and boundary layer dynamics in numerical models. Below, we provide a detailed discussion to clarify our experimental design and justify the vertical grid configuration in this study.

Our study aims to investigate differences in pollutant concentrations across different horizontal resolutions and explore how the turbulent mixing plays as a crucial role affecting pollutant concentrations at various horizontal resolutions. Therefore, we performed nested simulations using WRF-Chem's Ndwon method, where the three different resolutions refer to horizontal resolution, while the vertical resolution remains consistent across all experiments. The layer thicknesses in our boundary layer configuration are sufficient for capturing sub-kilometer-scale eddies and vertical mixing processes. Jiang et al. (2024) and Jiang and Hu (2023) have demonstrated that the number of model vertical layers primarily influences vertical distribution, with more vertical grid layers producing a more stable vertical structure under stable boundary conditions that can better resolve boundary layer turbulence. In our WRF-Chem downscaling simulations, the vertical grid structure was designed with a total of 50 terrain-following vertical eta-layers extending from the surface to the height of 15 km in all three resolution simulations, with 30 layers distributed below 2 km above the ground to describe the atmospheric boundary structure in detail. The vertical layer was strategically designed with 7 layers below 200 meters, each approximately 20 meters in height; 3 layers between 200 and 300 meters, each about 30 meters in height; 8 layers between 300 and 1000 meters, each approximately 80 meters in height, which is sufficient to explicitly resolve boundary layer turbulence.

That being said, we acknowledge that vertical grid resolution may play a more significant role in resolving eddies of different scales. Excessively coarse vertical grids (e.g., greater than 100 m layer thickness) could artificially suppress vertical mixing and misrepresent energy cascades across scales. When the vertical resolution is not sufficient to correctly resolve the vertical gradient, the chemical transport model usually fails to maintain the fine structure of the plume resulting in their over-dilution compared to observations (Eastham and Jacob, 2017; Zhuang et al., 2018). Studies confirm that high vertical resolution configurations substantially reduce system errors compared to lower resolution settings, making it suitable for both long-term average simulations and transient characteristics such as blocking events (Liu et al., 2022; Pelucchi et al., 2021; Teixeira et al., 2016; Tang et al., 2006; Roeckner et al., 2006).

While the vertical resolution is held constant in our study, we do recognize that it could influence the interpretation of the turbulence processes in certain scenarios, especially in regions with complex vertical structures. However, due to the already sufficient vertical resolution in our simulations, we focused on the effect of horizontal resolution on turbulence mixing, which remains the dominant factor in our experiments. While vertical resolution is important for resolving stratified flows and boundary layer instabilities, horizontal resolution predominantly controls representation of turbulent mixing in mesoscale

simulations when vertical mixing schemes are well-configured. We have added a brief discussion on the role of vertical resolution in the end of section Discussion of the revised manuscript: "Finally, while the vertical resolution is held constant in our study, we do recognize that it could influence the interpretation of the turbulence processes in certain scenarios, especially in regions with complex vertical structures. Therefore, future work should systematically explore the interplay between vertical resolution and pollutant concentration or aerosol-boundary layer feedbacks."

**Response:** Thank you for your constructive feedback. In this context, we are referring to the "horizontal resolution". We have updated the manuscript to clearly specify "horizontal resolution" in the Key Points section, as well as in all relevant parts of the manuscript where the resolution is mentioned. This clarification ensures that the resolution type is explicitly defined throughout the article.

●   *2. Line 418: what is the diurnal variation of BC emissions?*

**Response:** Thank you for your constructive feedback. The diurnal variation of BC emissions refers to the pattern in which emissions from all sectors peak during the daytime and are lower at night, as illustrated in Figure 1. The diurnal profiles of emissions from the five individual sectors (agriculture, industry, transport, energy, and residential) all show this same pattern. However, the diurnal variations are much stronger for agriculture, residential, and transportation sectors compared to industry and power plants (Du et al., 2020). We have added a brief introduction of the diurnal variation of BC emissions in section 3.2.1 of the revised manuscript: "The diurnal variation of BC emissions peak during the daytime and are lower at night."

[Figure]

Figure 1. Diurnal variation of the total BC emissions from five individual sectors (agriculture, industry, transport, energy, and residential).

Reference:

Du, Q., Zhao, C., Zhang, M., Dong, X., Chen, Y., Liu, Z., Hu, Z., Zhang, Q., Li, Y., Yuan, R., and Miao, S.: Modeling diurnal variation of surface PM2.5 concentrations over East China with WRF-Chem: impacts from boundary-layer mixing and anthropogenic emission, Atmospheric Chemistry and Physics, 20, 2839-2863, https://doi.org/10.5194/acp-20-2839-2020, 2020.

● *3. I noticed a few spelling errors, such as 'parametrized' (should be 'parameterized') on page 17, line 474. Please review the manuscript for similar issues.*

**Response:** Thank you for pointing out the spelling error. We have carefully reviewed the manuscript and corrected all similar issues, including the change from 'parametrized' to 'parameterized'. We appreciate your attention to detail and have ensured that the manuscript is free from such errors.

---

## Author Comment (AC4)

**Response Letter**

**Dear Editors and Reviewers:**

We sincerely thank you for facilitating the review process and providing valuable feedback. The comments received have been instrumental in enhancing our manuscript's quality and clarity.

We have carefully considered all the comments and have made comprehensive revisions to address each point raised. Our detailed responses to the reviewers' comments are provided below.

Yours sincerely,
Zining Yang and co-authors

**Reviewer #1**

*General comments:*

- *1. I would like to thank the authors for preparing a compelling response to my initial review, in which they clearly demonstrate that my technical concern regarding the parametrization of vertical mixing strength only applies to gases, but not to aerosols. I believe that the modelling community will benefit from articles that point out these hidden model aspects as clearly as possible, which this article now does.*

**Response:** Thank you for your positive feedback and for acknowledging the improvements made in the manuscript. We are glad that our response effectively addressed your technical concern regarding the parametrization of vertical mixing strength and clarified that it applies to gases but not to aerosols. We also appreciate your recognition of the value of pointing out such model aspects clearly, and we are pleased that the revised version of the article meets that expectation.

- *2. I have two more minor comments, and a few proposals for language improvements.*

**Response:** Thank you for your additional comments and suggestions. We appreciate your attention to detail and will carefully consider the minor comments and language improvement proposals. The necessary revisions will be made to enhance the clarity and quality of the manuscript.

*Minor comments:*

- *1. I still wonder how the findings of this article should be interpreted in the context of Du et al. (2021). Figure 7 shows a range of nighttime mixing coefficients in the range of 0-3 m²/s. As pointed out by the authors, the clipping of mixing strength does not apply to black carbon (BC), but Du et al. (2021) have nonetheless shown that perhaps it should, because their clipping at 5 m²/s resulted in significantly more realistic surface BC. There is also no reason why trace gases should undergo a different mixing routine than aerosols. Based on these two arguments, it seems plausible that the exclusion of aerosols from the clipping procedure was an oversight. The authors need not concern with this question directly, but they should attempt to answer the following (perhaps in the discussion of Figure 7)::*
*Assuming it were appropriate to apply the clipping also to the mixing of aerosols, would this not potentially change the findings of this article? After all, the entire variance of the nighttime mixing values would vanish if the values were clipped to 5 m²/s as suggested by Du et al. (2021). At daytime, where mixing coefficients are higher anyways, this logic would not apply.*

**Response:** Thank you for your thoughtful comment and for bringing up the comparison with Du et al. (2021). We acknowledge the potential influence of clipping the mixing coefficients on the simulation results, especially with respect to the surface black

carbon (BC) concentrations as demonstrated by Du et al. (2021). However, as we note in our response, the use of specific thresholds, such as the 5 m²/s threshold proposed by Du et al., is primarily empirical in nature. These thresholds are designed to compensate for missing physical processes within the model, often by artificially enhancing mixing intensity. While such approaches may yield improvements in simulation results, they do not necessarily reflect the underlying physical mechanisms at play. Our study aims to understand the root causes of the underestimation of nighttime mixing intensity in the model, with a particular focus on the role of model resolution in turbulent mixing processes. Instead of relying on empirical thresholds to match the model output to observed data, we seek to address the fundamental physical processes that lead to discrepancies in mixing strength. We believe that threshold-based methods, while useful for aligning simulations with observations, do not provide a sufficient theoretical basis and may obscure a deeper understanding of the physical dynamics involved. Therefore, while we do not dispute the value of threshold adjustments in improving model performance, our approach deliberately avoids these artificial modifications in favor of investigating the underlying mechanisms that affect nighttime mixing intensity. In this context, we do not apply the clipping approach to aerosols in our analysis, as we believe it would detract from our goal of improving the model's representation of the physical processes driving turbulent mixing. We hope this clarifies our position and addresses the concern raised regarding the potential impact of applying clipping to aerosol mixing coefficients.

In the revised manuscript (Section 2.2.2), we have added these detailed explanation: "The publicly available version of WRF-Chem defines a default lower limit of 0.1 m²/s for particulate matter mixing coefficients. We did not implement the adjustment proposed by Du et al. (2020), which suggest raising the lower limit of PBL mixing coefficient from 0.1 $m^2$/s to 5 $m^2$/s within the PBL. Although setting specific thresholds can improve simulation results, such thresholds are predominantly empirical in nature, whether based on CO and $PM_{2.5}$ emissions or the 5 m²/s threshold suggested by Du et al. (2020). These threshold adjustments effectively compensate for missing physical processes in the model by artificially enhancing mixing intensity. Our approach focuses on understanding the physical mechanisms responsible for the model's underestimation of nighttime mixing intensity, with particular emphasis on how model resolution affects turbulent mixing processes. Rather than employing empirical thresholds to align model output with observations, we aim to investigate the fundamental causes of the discrepancies. We contend that threshold approaches rely heavily on empirical data, lack sufficient theoretical foundation, and may impede comprehensive understanding of the underlying physical processes. Consequently, this study utilizes the default particulate matter turbulent mixing coefficients in the model for our analyses."

In the revised manuscript (Section 3.2.2), we have also added another detailed explanation:

"We contend that threshold approaches are primarily based on empirical data and may impede comprehensive understanding of the underlying physical processes. In our study, particulate matter mixing coefficients are directly calculated through boundary layer parameterization without adjustments based on empirical settings."

- *2. The authors discuss the correlation between surface types / land usage and mixing strength. In doing so, they mainly*

*focus on how turbulence is affected by surface roughness. But there is another major mechanism to be discussed: The differences in radiative uptake among different surface types. Some surfaces absorb more of the actinic flux, and subsequently transfer this energy through sensible heat to the air above, which significantly contributes to the day-time convective boundary layer.*

**Response:** Thank you for your detailed review and constructive comments on our manuscript. You highlighted an important aspect regarding the differences in radiative uptake among different surface types and their impact on the convective boundary layer formation, which we fully agree is a crucial mechanism. In response to your suggestion, we have expanded the discussion in Section 4 to include this important mechanism. The added text thoroughly addresses how different surface types affect the convective boundary layer (CBL) and turbulence mixing strength through variations in radiative flux absorption, reflection, and heat exchange. We discuss how urban areas with lower albedo absorb more shortwave radiation, leading to increased surface temperature and energy transfer to the atmosphere through sensible heat. In contrast, we explain how vegetated areas with higher albedo release more latent heat through transpiration while reducing sensible heat output. Furthermore, we elaborate on how the balance between sensible and latent heat fluxes across different surface types impacts turbulence intensity and CBL depth. For instance, we note that urban areas with stronger sensible heat flux generate more intense thermal convection, while vegetated areas dominated by latent heat flux may develop more stable atmospheric conditions. We conclude by emphasizing that future studies on land use impacts on turbulence mixing should consider not only surface roughness but also radiative flux differences, heat exchange mechanisms, and the comprehensive effects of surface albedo on turbulence development. In the revised manuscript (Section 4), we have added this detailed explanation: "Moreover, in addition to the influence of surface roughness on turbulence intensity, surface type significantly affects the CBL and turbulence mixing strength through differences in radiative flux absorption, reflection, and heat exchange. There are substantial variations in the absorption and reflection of shortwave radiation across different surface types. Urban areas typically have lower albedo, absorbing more shortwave radiation, which increases surface temperature and transfers energy to the atmosphere as sensible heat. In contrast, vegetated areas generally have higher albedo and, through transpiration, release more latent heat while reducing sensible heat output. These differences in energy exchange between surface and atmosphere directly influence turbulence strength. Furthermore, the varying balance between sensible and latent heat fluxes across different surface types impacts turbulence intensity and CBL depth. For instance, urban areas, with stronger sensible heat flux, tend to generate more intense thermal convection, often resulting in a shallower CBL, while vegetated areas, with predominant latent heat flux, may develop more stable atmospheric conditions, potentially leading to a deeper CBL with weaker turbulence. These mechanisms of radiative absorption and heat exchange are crucial in the formation of the diurnal CBL and determining turbulence intensity. Future studies on land use impacts on turbulence mixing should therefore consider not only surface roughness but also radiative flux differences, sensible and latent heat exchange mechanisms, and the comprehensive effects of surface albedo on turbulence development."

*Comments on language / writing style:*

- *1. 101: change to "Previous research has indicated (...)". The use of the plural "researches" appears in other parts of the article as well and appears false to me.*

**Response:** Thank you for your valuable comment. I have carefully revised the manuscript as per your suggestion. Specifically, I have changed the plural form "researches" to the singular "research" throughout the manuscript, including the part you mentioned. I appreciate your attention to this detail.

- *2. 101-106: This research summary is only helpful if the readers are made aware of the differences between the model code in the presented article and the referenced literature. For example, the study by Kuhlmann et al. (2003) is over 20 years old, and uses a different model at far lower resolutions. I would argue that such literature has limited relevance for the article's research question.*

**Response:** Thank you for your valuable comment. We understand your concern regarding the relevance of the literature referenced, particularly the study by Kuhlmann et al. (2003). As you pointed out, this study is over 20 years old and uses a different model at much lower resolutions, which may indeed limit its direct relevance to the research question presented in our article. In response to your feedback, we have removed references to outdated or less relevant studies, including Kuhlmann et al. (2003), to ensure that the literature cited is more aligned with the current modeling approaches, higher resolutions, and research objectives. We agree that it is crucial to reference studies that better match the scope and methods of our work, and we will now focus on more recent studies that are directly relevant to the model framework, resolution, and physical processes discussed in our paper. In the revised manuscript, we will explicitly clarify the differences between the model code used in this article and the referenced literature, especially in terms of model selection, resolution, and the physical processes involved, to avoid any potential confusion. We believe these revisions will enhance the relevance and scientific rigor of the paper.

- *3. 163: Here and in other places, the term "multi-resolutions" is used, which I would recommend to replace by "multiple resolutions", "different resolutions", or "various resolutions".*

**Response:** Thank you for your suggestion regarding the use of the term "multi-resolutions." While we understand that terms like "multiple resolutions," "different resolutions," or "various resolutions" are commonly used, we have intentionally chosen the term "multi-resolutions" to more precisely convey the concept of utilizing several distinct resolution levels in our analysis. We believe this term captures the intended meaning more effectively in the context of our study. However, we appreciate your input and will consider your suggestion carefully as we finalize the manuscript.

- *4. 177: The sentence with "contains some treatments" is not very informative. The authors could write instead: "WRF-Chem treats photochemistry of trace gases and aerosol-related processes with various different schemes (e.g. the Statewide Air Pollution Research Center (SAPRC99) photochemical mechanism and the Model for Simulating Aerosol Interactions and Chemistry (MOSAIC))."*

**Response:** Thank you for your constructive comment. Following your suggestion, we have revised the sentence to provide more specific and informative details. The revised sentence now reads: "WRF-Chem treats photochemistry of trace gases and aerosol-related processes with various different schemes (e.g., the Statewide Air Pollution Research Center (SAPRC99) photochemical mechanism and the Model for Simulating Aerosol Interactions and Chemistry (MOSAIC))." We believe this revision enhances the clarity of the manuscript.

- *5. 190: remove "also".*

**Response:** Thank you for your suggestion. We have removed the word "also" as per your recommendation. We have updated Section 2.1.1 of the manuscript to include this modification:" The SAPRC99 photochemical mechanism (Carter, 2000) is chosen to simulate the gas-phase chemistry, and the MOSAIC is selected for aerosol processes (Zaveri and Peters, 1999; Zaveri et al., 2008). "

- *6. 313: Split into two sentences: "This study primarily focuses on BC. The spatial distribution of BC emissions is shown in Figure 22."*

**Response:** Thank you for your suggestion. We have split the sentence into two as per your recommendation: "This study primarily focuses on BC. The spatial distribution of BC emissions is shown in Figure 2."

- *7. 322: Is Zhang et al. (2021) a good reference to the MEGAN model? As far as I know, MEGAN is usually cited as: Guenther, A., Karl, T., Harley, P., Wiedinmyer, C., Palmer, P. I., and Geron, C.: Estimates of global terrestrial isoprene emissions using MEGAN (Model of Emissions of Gases and Aerosols from Nature), Atmos. Chem. Phys., 6, 3181–3210, https://doi.org/10.5194/acp-6-3181-2006, 2006.*
  *This refers to the original version of MEGAN, and newer versions have been published. Regardless, Guenther et al. should be cited in the context of MEGAN.*

**Response:** Thank you for your valuable feedback. We acknowledge the concern regarding the citation of Zhang et al. (2021) in relation to the MEGAN model. As you correctly pointed out, the appropriate reference for the MEGAN model is Guenther et al. (2006), which describes the original version of the model. We have revised the manuscript to include Guenther et al. (2006) as the primary reference for MEGAN and have adjusted the citation of Zhang et al. (2021) accordingly. Additionally, if newer versions of the MEGAN model are referenced in the manuscript, we will ensure that the relevant citations for those

updates are properly included. In the revised manuscript (Section 4), we have added this cite: "Biogenic emissions were calculated using the Model of Emissions of Gases and Aerosols from Nature (MEGAN) v3.0 model (Guenther, 2007; Zhang et al., 2021)."

Reference:

Guenther, A.: Estimates of global terrestrial isoprene emissions using MEGAN (Model of Emissions of Gases and Aerosols from Nature) (vol 6, pg 3181, 2006), Atmospheric Chemistry and Physics, 7, 4327-4327, https://doi.org/10.5194/acp-7-4327-2007, 2007.

- *8. 350: Change "derived" to "used".*

**Response:** Thank you for your suggestion. We have made the recommended change and revised the sentence to "We used the hourly BC observations from the air quality monitoring site on the campus of USTC during spring (March 10 to March 20, 2019)." We appreciate your input, which has improved the clarity of the text.

- *9. 391: "We believe this approach (...)" should be rephrased to a more objective statement.*

**Response:** Thank you for your constructive feedback. We have revised the sentence as suggested, changing "We believe this approach (...)" to "This approach effectively captures daily variability patterns without losing essential detail." We also ensured that the revised text maintains clarity and objectivity. Your suggestion has improved the precision of the statement.

- *10. 489: Change to "Figure S8b demonstrates (...)".*

**Response:** Thank you for your suggestion. We have made the recommended change and revised the sentence to "Figure S8b demonstrates (...)." Your input has helped improve the clarity and accuracy of the text.

- *11. 647: It is unclear to me what "intension" means in this context.*

**Response:** Thank you for your suggestion. We have removed the term "intension," as it was unclear in this context, and revised the sentence to: "Figure S13 shows that the turbulent mixing coefficient parameterized at 5 km resolution is larger than that at 1 km resolution, which fails to explain the similar surface concentrations in these two higher-resolution (5 km and 1 km) simulations." We believe this revision clarifies the intended meaning.

- *12. 671: Remove "vertically upward".*

**Response:** Thank you for your suggestion. We have removed the phrase "vertically upward" as per your recommendation. This revision has helped improve the clarity of the text.

- *13. 863: Change to "dynamical framework of the model".*

**Response:** Thank you for your suggestion. We have made the recommended change and revised the text to "dynamical

framework of the model." This adjustment has improved the accuracy and clarity of the statement.

- *14. 865: Change to "The resolution of dynamic processes reduces (...)".*

**Response:** Thank you for your valuable feedback. We have revised the sentence as suggested, changing it to "The resolution of dynamic processes reduces reliance on traditional parameterization schemes, thereby decreasing the PBL mixing coefficient parameterized at finer resolutions." This modification has enhanced the clarity and precision of the statement.